# Individual differences in tail risk sensitive exploration using Bayes-adaptive Markov decision processes

**Tingke Shen\*, Peter Dayan**

Max Planck Institute for Biological Cybernetics, Tübingen, Germany

## eLife Assessment

Shen et al. present a computational account of individual differences in mouse exploration when faced with a novel object in an open field from a previously published study (Akiti et al.) that relates subject-specific intrinsic exploration and caution about potential hazards to the spectrum of behaviors observed in this setting. Overall, this computational study is an **important** contribution that leverages a very general modeling framework (a Bayes Adaptive Markov Decision Process) to quantify and interrogate distinct drivers of exploratory behavior under potential threat. Given their assumptions, the modeling results are **convincing**: the authors are able to describe a substantial amount of the behavioral features and idiosyncracies in this dataset, and their model affords a normative interpretation related to inherent risk aversion and predation hazard "flexibility" of individual animals and should be of broad interest to researchers working to understand open-ended exploratory behaviors.

**\*For correspondence:**
kshentingke@gmail.com

**Competing interest:** The authors declare that no competing interests exist.

**Abstract** Novelty is a double-edged sword for agents and animals alike: they might benefit from untapped resources or face unexpected costs or dangers such as predation. The conventional exploration/exploitation tradeoff is thus colored by risk sensitivity. A wealth of experiments has shown how animals solve this dilemma, for example, using intermittent approach. However, there are large individual differences in the nature of approach, and modeling has yet to elucidate how this might be based on animals' differing prior expectations about reward and threat, and differing degrees of risk aversion. To capture these factors, we built a Bayes-adaptive Markov decision process model with three key components: an adaptive hazard function capturing potential predation, an intrinsic reward function providing the urge to explore, and a conditional value at risk (CVaR) objective, which is a contemporary measure of trait risk sensitivity. We fit this model to a coarse-grain abstraction of the behavior of 26 animals who freely explored a novel object in an open-field arena. We show that the model captures both quantitative (frequency, duration of exploratory bouts) and qualitative (with distinguished, cautious, tail-behind approach) features of behavior, including the substantial idiosyncrasies that were observed. Some animals begin with cautious exploration and quickly transition to a confident approach to maximize exploration for reward; we classify them as potentially more risk neutral and enjoying a flexible hazard prior. By contrast, other animals only ever approach in a cautious manner and display a form of self-censoring; they are characterized by potential risk aversion and high and inflexible hazard priors. Explaining risk-sensitive exploration using factorized parameters of reinforcement learning models could aid in the understanding, diagnosis, and treatment of psychiatric abnormalities such as anxiety disorders.

## Introduction

In naturalistic environments, novelty can be a source of both reward and dangers. Despite these dueling aspects, investigations of novelty in reinforcement learning have mostly focused on neophilia driven by optimism in the face of uncertainty, and so information-seeking (*Duff, 2002b*; *Dayan and Sejnowski, 1996*; *Gottlieb et al., 2013*; *Wilson et al., 2014*). Neophobia has attracted fewer computational studies, apart from some interesting evolutionary analyses (*Greggor et al., 2015*).

Excessive novelty seeking and excessive novelty avoidance can both be maladaptive – they are flip sides of a disturbed balance. Here, we seek to examine potential sources of such disturbances, for instance, in distorted priors about the magnitude or probabilities of rewards (which have been linked to mania; *Radulescu and Niv, 2019*; *Bennett and Niv, 2020*; *Eldar et al., 2016*) or threats (linked to anxiety and depression; *Bishop and Gagne, 2018*; *Paulus and Yu, 2012*), or in extreme risk attitudes (*Gagne and Dayan, 2022*).

To do this, we take advantage of a recent study by *Akiti et al., 2022* on the behavior of mice exploring a familiar open-field arena after the introduction of a novel object near to one corner. The mice could move freely and interact with the object at will. *Akiti et al., 2022* performed detailed analyses of how individual animals' trajectories reflected the novel object, including using DeepLabCut (*Mathis et al., 2018*) to track the orientation of the mice relative to the object and MOSEQ (*Wiltschko et al., 2020*) to extract behavioral 'syllables' whose prevalence was affected by it. The animals differed markedly in how they approached the object and in what pattern. For the former, *Akiti et al., 2022* observed two characteristic positionings of the animals when near to the object: 'tail-behind' (bouts where the animal's nose was closer to the object than the tail for the entire bout) and 'tail-exposed' (bouts where the animal's tail is closer to the object than the nose at some point during the bout), associated, respectively, with cautious risk assessment and engagement. For the latter, there was substantial heterogeneity, with all animals initially performing tail-behind approach, but some taking much longer (or failing altogether) to transition to tail-exposed approach.

*Akiti et al., 2022* provide a model-free reinforcement learning account of their data, focusing on the prediction of threat and its realization in the tail of the striatum (TS). Here, we provide a model-based reinforcement learning account, focusing on the rich details of the dynamics of approach carefully characterized by *Akiti et al., 2022*. These include intermittency (i.e. why animals retreat from the object), approach drive (or why animals approach in the first place), the significant long-run approach of cautious animals despite having reached periods of 'avoidance' behavior, and how the intensity of approach increases when the other animals transition from risk assessment to engagement and then decreases in the long run of the 'engagement' phase. Our model provides an alternative explanation for why animals learn to avoid the novel object in a completely benign environment. Through modeling these additional statistics and behaviors, we reveal the multidimensional aspect of caution in exploration that cannot be captured just in terms of time spent at the object.

We model an abstract depiction of the behavior of individual mice by combining the Bayesadaptive Markov decision process (BAMDP) treatment of rational exploration (*Dearden et al., 2013*; *Duff, 2002b*; *Guez et al., 2013*) with two sources of risk sensitivity: the prior over the potential hazard associated with the object, and the conditional value at risk (CVaR) probability distortion mechanism (*Artzner et al., 1999*; *Chow et al., 2015*; *Gagne and Dayan, 2022*; *Bellemare et al., 2023*).

In a BAMDP, the agent maintains a belief about the possible rewards, costs, and transitions in the environment and decides upon optimal actions based on these beliefs. Since the agent can optionally reuse or abandon incompletely known actions based on what it discovers about them, these actions traditionally enjoy an exploration bonus or 'value of information', which generalizes the famous Gittins indices (*Gittins, 1979*; *Weber, 1992*). In addition to beliefs about reward, the agent also maintains a belief about potential hazard, which is the first source of risk sensitivity. These beliefs are initialized as prior expectations about the environment and so are readily subject to individual differences.

In addition to beliefs about hazards which may be specific to a particular environment, we include a second, potentially more general, source of risk sensitivity. That is, we consider optimizing the CVaR, in which agents concentrate on the average value within lower (risk-averse) or upper (risk-seeking) quantiles of the distribution of potential outcomes (*Rigter et al., 2021*). In the context of a BAMDP, this can force agents to pay particular attention to hazards. More extreme quantiles are associated

with more extreme risk sensitivity and again are a potential locus of individual differences (as examined in regular Markov decision processes in the context of anxiety disorders in *Gagne and Dayan, 2022*).

In sum, we present a behavioral model of risk-sensitive exploration, with an agent computing optimal actions using the BAMDP framework under a CVaR objective. This model provides a normative explanation of individual variability – the agent makes decisions by trading off potential reward and threat in a principled way. Different priors and risk sensitivities lead to exploratory schedules that differ in duration, frequency, and type of approach (risk assessment versus engagement) through time. We report features of the different behavioral trajectories the model is able to capture, providing mechanistic insight into how the trade-off between potential reward and threat leads to rational exploratory schedules. Behavioral phenotypes emerge from the interaction of the separate computational mechanisms elucidated by our model-based treatment. This paves the way for future experimental investigations of these mechanisms, including the unexpected non-identifiability of our two sources of risk sensitivity: hazard priors and CVaR.

## Results

### Behavior phases and animal groups

Our goal is to provide a computational account of the exploratory behavior of individual mice under the assumption that they have different prior expectations and risk sensitivities. We start from *Akiti et al., 2022*'s observation that the animal approaches and remains within a threshold distance (determined by them to be 7 cm) of the object in 'bouts' which can be characterized as 'cautious' or tail-behind (if the animal's nose lies between the object and tail) or otherwise 'confident' or tail-exposed. We sought to capture these qualitative differences (cautious versus confident) as well as aspects of the quantitative changes in bout durations and frequencies as the animal learns about their environment.

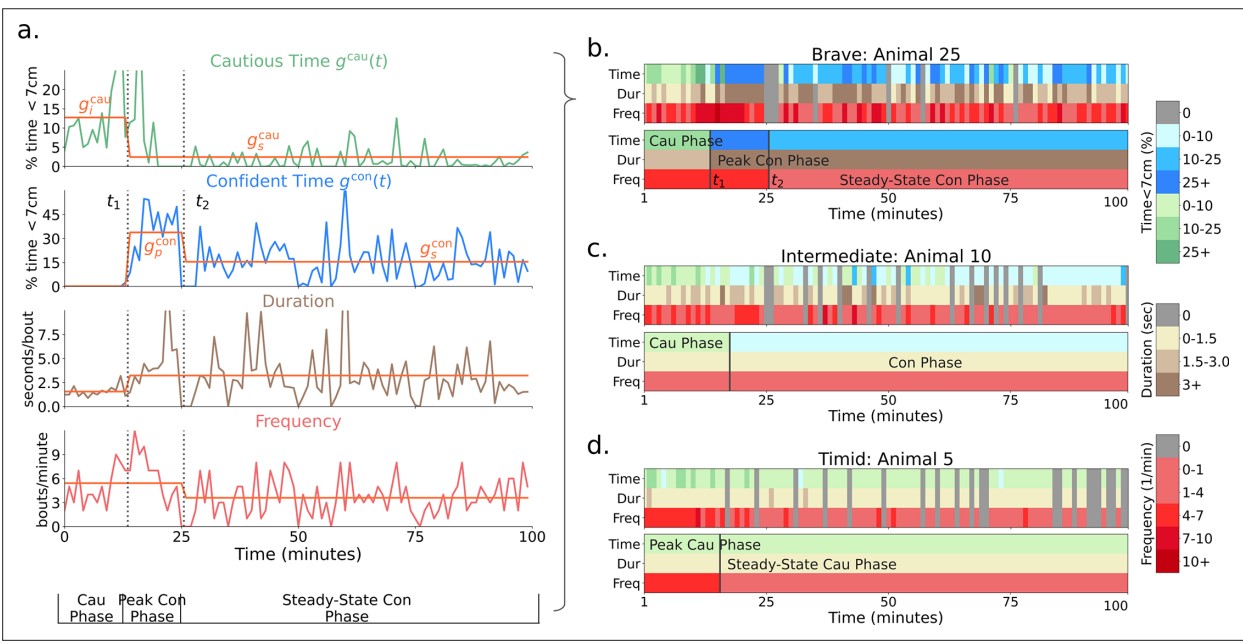

**Figure 1.** Extraction of phase-averaged behavioral statistics from minute-resolution bouts data. (**a**) Detailed visualization of minute-to-minute statistics of animal 25 (in the sessions after the introduction of the novel object). From top to bottom, the plots show % time within (*Akiti et al., 2022*'s 7 cm threshold of the object with (cautious) and without (confident) tail-behind, the length of a bout at the object and the number of bouts per minute. Orange lines are the box-car functions fitted to segment phases and illustrate the change in time, duration, and frequency statistics across phases. The transition points $t_1$ and $t_2$ as well as the initial cautious $g_i^{cau}$, final cautious $g_s^{cau}$, peak confident $g_p^{con}$ and steady-state confident $g_s^{con}$ approach percentage times are shown. The right plots show examples of minute-to-minute and phase-averaged approach time, duration, and frequency for (**b**) brave, (**c**) intermediate, and (**d**) timid animals. Note that animals are ordered by the group-timidity animal index (see A spectrum of risk-sensitive exploration trajectories). Green indicates cautious and blue indicates confident approach. Darker colors indicate higher values. For the purpose of modeling, we average the idiosyncrasies of behavior over phases and thereby characterize a high-level summary of learning dynamics.

To make this readily possible, we abstracted the data in two ways: averaging bout statistics over time and clustering the animals into three groups with operationally distinct behaviors.

*Akiti et al., 2022*'s classification into bouts can be seen as a very useful abstraction over some of the detailed complexities of the behavior. In order to focus narrowly on interaction with the object, we abstracted further. In particular, instead of modeling the details of the animals' spatial interaction with the object, we fitted boxcar functions to the percentages of its time $g^{cau}(t), g^{con}(t)$ that the animal spends in cautious and confident bouts around time $t$ in the apparatus. We can then well encompass the behavior of most animals via four coarse phases of behavior that arise from two binary factors: whether the animal is mainly performing cautious or confident approaches, and whether bouts happen frequently, at a peak rate, or at a lower, steady-state rate. The time an animal spends near the object in one of these phases reflects the product of how frequently it visits the object and how long it stays per visit. We average these two factors within each phase.

Consider the behavior of the animal in *Figure 1a*. Here, $g^{cau}(t)$ (top graph) makes a transition from an initial level $g_i^{cau}$ (during the 'cautious' phase) to a final steady-state level $g_s^{cau}$ (which we simplify as being $g_s^{cau} = 0$) at a transition point $t = t_1$. At the same time point, $g^{con}(t)$ (second row) makes a transition from 0 to a peak level $g_p^{con}$ of confident approach (defining the 'peak confident' phase). Finally, there is another transition at time $t_2$ from peak to a steady-state confident approach time $g_s^{con}$ (in the 'steady-state confident' phase). The lower two rows of *Figure 1a* show the duration of the bouts in the relevant phases and the frequency per unit time of such bouts. The upper panel of *Figure 1b* shows the same data in a more convenient manner. The colors in the top row indicate the type of approach (green is cautious; blue is confident). The second and third rows indicate the duration and frequency of approach. Darker colors represent higher values.

The orange-colored lines in *Figure 1a* and the lower panel in *Figure 1b* render the abstracted behavior of this animal in an integrated form, showing how we generate 'phase-level' statistics from minute-to-minute statistics. Averaging statistics over phases ignores idiosyncrasies of behavior and allows us to fit the high-level statistics of behavior: phase-transition times, phase-averaged durations, and frequencies. We consider animal 25 to be a 'brave' animal because of its transition to peak and then steady-state confident approach. There were 12 brave mice out of the 26 in total.

*Figure 1c* shows an example of another characteristic 'intermediate' animal. This animal makes a transition from cautious to confident approach (where both duration and frequency of visits can change), but the approach time during the confident phase $g_s^{con}$ does not decrease. Hence, intermediate animals do not have a transition from peak to steady-state confident phase. There were five such intermediate mice.

*Figure 1d* shows the behavior of an example of the last class of 'timid' animals. This animal never makes a transition to confident approach. Hence, for it, $g^{con}(t) = 0$. However, the cautious approach time makes a transition to a non-zero steady state ($g_s^{cau} > 0$), often via a change in frequency, defining the fourth phase (steady-state cautious). There were nine such timid mice.

*Figure 2* summarizes our categorization of the animals into the three groups: brave, intermediate, and timid based on the phases identified in the animal's exploratory trajectories. Timid animals spend no time in confident approach and are plotted in orange at the origin of *Figure 2*. Brave animals differ from intermediate animals in that their approach time during the first 10 min of the confident phase is greater than the last 10 min (steady-state phase). Brave animals are plotted in green above and intermediate animals are plotted in black below the $y = 1$ line in *Figure 2*.

## A Bayes-adaptive model-based model for exploration and timidity

### State description

We use a model-based Bayes-adaptive reinforcement learning model (BAMDP) to provide a mechanistic account of the behavior of the mice under threat of predation. This extends the model-free description of threat in *Akiti et al., 2022* by constructing various mechanisms to explain additional facets of the dynamics of the behavior.

Underlying the BAMDP is a standard multi-step decision-making problem of the sort that is the focus of a huge wealth of studies (*Russell and Norvig, 2016*). We cartoon the problem with the four real and four counterfactual states shown in *Figure 3*. The nest is a place of safety, modeling all places in the environment away from the object, ignoring, for instance, the change to thigmotactic behavior that the mice exhibit when the object is introduced. The animal can choose to stay at the

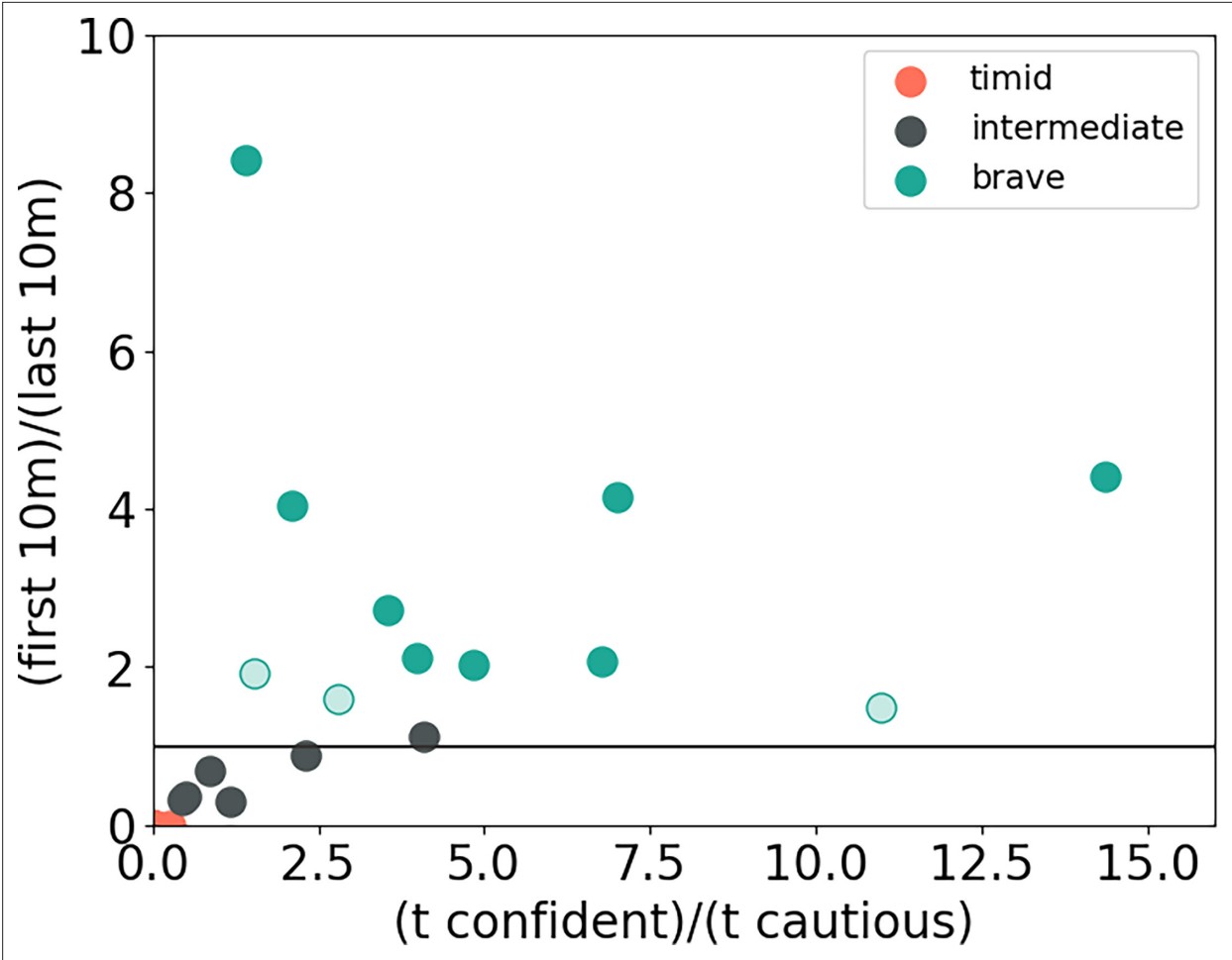

**Figure 2.** Categorization of animals into timid, intermediate, and brave groups based on cautious and confident bout statistics. The *x*-axis shows the ratio of total time spent in confident versus cautious bouts. The *y*-axis shows the ratio of bout time in the first 10 min of confident approach and the last 10 min of confident approach (set to 0 for timid animals that do not have a confident phase). The horizontal line indicates $y = 1.0$. All nine timid animals are close to the origin. We separate brave and intermediate animals according to the $y = 1$ line. Solid green dots are brave animals that pass the Benjamini–Hochberg procedure for $y > 1$ at level $q = 0.05$ according to a random permutation test. Hollow dots represent brave animals that did not pass. We decided to model these animals as brave since they had $y > 1.5$ and hence a relatively clear confident-peak to confident-steady-state transition point. Modeling them as intermediate animals instead would not have significantly affected our results. Black dots are intermediate animals. They did not pass the Benjamini–Hochberg procedure for either $y > 1$ or $y < 1$.

nest (possibly for multiple steps) or choose to approach the object. For convenience, we adopt *Akiti et al., 2022*'s binary classification of approach, while acknowledging the substantial simplification this classification entails (a fuller but more complex characterization of approach would be continuous and multidimensional). We represent the tail-behind (respectively, tail-exposed) approach as transitioning to the cautious (respectively, confident) object state.

At an approach state, the modeled agent can either stay or return to the nest via the retreat state; the latter happens anyhow after four steps. The animal also imagines the (in reality, counterfactual) possibility of being detected by a potential predator. It can then either manage to escape back to the nest or alternatively expire. We parameterize costs associated with the various movements and also the probability of unsuccessful escape starting from confident ($p_1$) or cautious ($p_2 < p_1$) approach.

We describe the dilemma between a cautious and a confident approach as a calculation of the risk and reward trade-off between the two types of approaches. A cautious approach (the 'cautious object' state) has a lower (informational) reward (e.g. because in the cautious state the animal spends more cognitive effort monitoring for lurking predators rather than exploring the object). However, a cautious approach leads to a lower probability of expiring if detected than does a confident approach

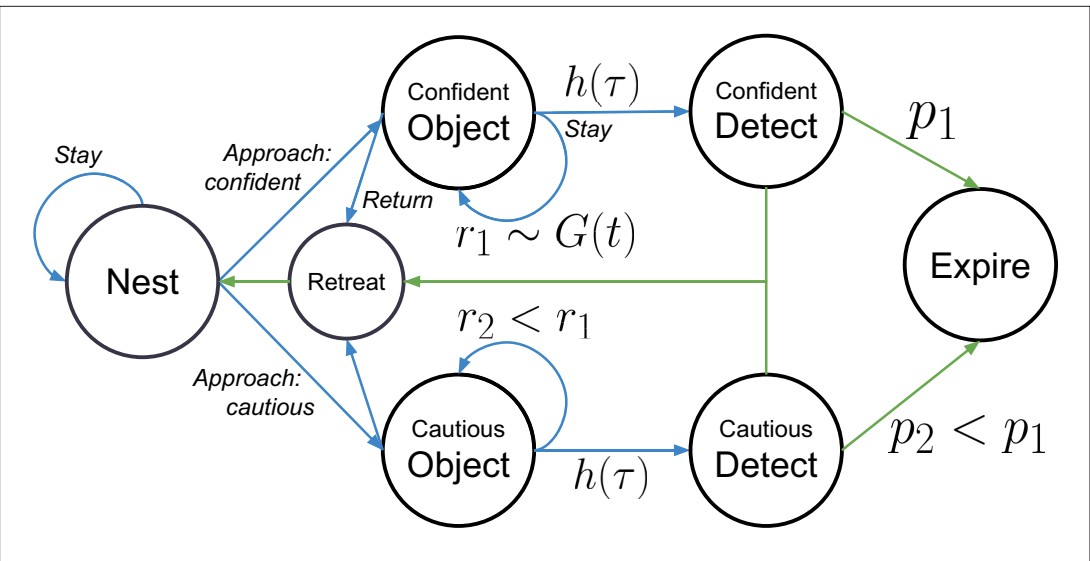

**Figure 3.** Markov decision process underlying the Bayes-adaptive Markov decision process (BAMDP) model. Four real (nest, cautious object, confident object, retreat) and three imagined (cautious detect, confident detect, dead) states. Agent actions are italicized. Blue arrows indicate (possibly stochastic) transitions caused by agent actions. Green arrows indicate (possibly stochastic) forced transitions. A cautious approach provides less informational reward $r_2 < r_1$ but has a smaller chance of death $p_2 < p_1$ compared to a confident approach. Travel and dying costs are not shown.

(the 'cautious object' state) (e.g. because in the cautious state the animal is better poised to escape). Risk aversion modulates the agent's choice of approach type.

The next sections describe the components of the BAMDP model: a characterization of the time-dependent risk of predation, an informational reward for exploration, and a method for handling risk

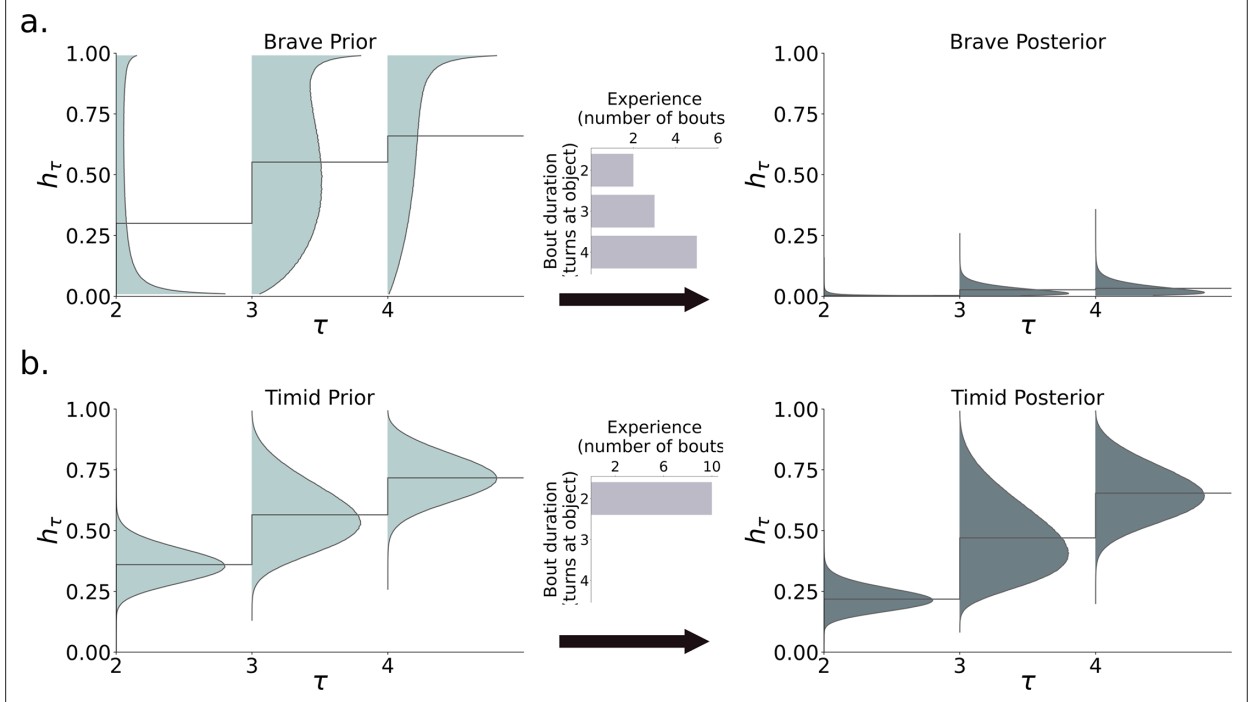

**Figure 4.** Hazard function learning for (**a**) brave and (**b**) timid animals. Brave animals start with a flexible hazard prior with a low mean for $h_2$. This leads to longer bouts (first length 2, then 3 and 4), which imply that the hazard posterior quickly approaches zero (here, after 10 bouts). Timid animals start with an inflexible hazard prior with a higher mean $h_2$, and are limited to length bouts. The hazard posterior only changes slightly after 10 bouts.

sensitivity. Finally, we will discuss the way we fitted individual mice and present a full analysis of their behavior. We report on recovery simulations in the supplement.

## Modeling threat with a Bayesian, generalizing hazard function

While exploring the novel object in the 'object' state, the decision problem allows for the possibility of detection, and then attack, by a predator whose appearance is governed by a temporal hazard function (see *Figure 4*).

Formally, the probability of detection given either cautious or confident approach is modeled using the hazard function $h_\tau$, where $\tau$ is the number of steps the animal has so far spent at the object in the current bout. In a key simplification, this probability resets back to baseline upon a return to the nest. We treat the hazard function as being learned in a Bayesian manner, from the experience (in this case, of not being detected). We assume that the animal has the inductive bias that the hazard function is increasing over time, reflecting a potential predator's evidence accumulation process about the prey. Therefore, we derive it from a succession of independent Beta-distributed random variables $\theta_1 = 0; \theta_\tau \sim \mathrm{Beta}(\mu_\tau, \sigma_\tau), \tau > 1$ as

$$h_\tau = 1 - \prod_{j=1}^{\tau} (1 - \theta_j) \tag{1}$$

$$= h_{\tau-1} + (1 - h_{\tau-1}) \theta_\tau, \qquad \text{for } \tau > 1 \tag{2}$$

rather as in what is known as a stick-breaking process. Note that, for convenience, we parameterize the Beta distribution in terms of its mean $\mu$ and standard deviation $\sigma$ rather than its pseudocounts, as is perhaps more common.

*Equation 2* shows that the hazard function is always increasing. As we will see, the duration of bouts at the object depends on the (discrete) slope of the hazard function, with steep hazard functions leading to short bouts. In our model, the agent can stay at the object 2, 3, or 4 turns (we take $\theta_1 = 0$ as a way of coding actual approach).We therefore sometimes refer to cautious –$k$ or confident–$k$ bouts in which the model animal spends $k = \{2, 3, 4\}$ steps at the object. Hence, the collection of random variables, $h_\tau$, is derived from six parameters (the mean $\mu_\tau$ and the standard deviation $\sigma_\tau$ of the Beta distribution for the turn). These start at initial prior values and are subject to an update from experience. Here, that experience is exclusively negative, since there is no actual predator; this implies that the update has a simple, closed form (see Methods). The animals' initial ignorance, which is mitigated by learning, makes the problem a BAMDP, whose solution is a risk-averse itinerant policy.

A particular characteristic of the noisy-or hazard function of *Equation 1* is that the derived bout duration increases progressively. This is because not being detected at $\tau = 2$, say, provides information that $\theta_2$ is small, and so reduces the hazard function for longer bouts $\tau > 2$.

*Figure 4* shows the fitted priors of a brave (top) and timid (bottom) animal, as well as the posteriors after ten exploratory bouts. The brave animal starts with a high variance prior. This flexibility allows it to transition from short, cautious bouts (duration $\tau = 2$) to longer confident bouts (duration $\tau = 3, 4$), reducing the hazard function to near zero. The timid animal has a low variance prior and does not stay long enough at the object to build sufficient confidence (only performing duration $\tau = 2$ bouts). As a result, its posterior hazard function remains similar to its prior.

## Modeling the motivation to approach

We model the mouse's drive to approach the object as stemming from its belief that the object might be rewarding. In a fully Bayesian treatment, the agent would maintain a posterior over the possibility of rewards and would enjoy a conventional, informational, Bayes-adaptive exploration bonus encouraging it to approach the object. However, this would add substantial computational complexity. Thus, instead, we use a simple, heuristic, exploration bonus $G(t)$ (*Kakade and Dayan, 2002*). The model mouse moves from the 'nest' state to the 'object' state when this exploration bonus exceeds the costs implied by the risk of being attacked.

We characterize the exploration bonus as coming from an initial 'pool' $G_0$ that becomes depleted when the animal is at the object, as it experiences a lack of reward, but is replenished at a steady rate $f$ when the animal is at the nest, through forgetting or potential change. We model the animal as harvesting this exploration bonus pool more quickly under confident than cautious approaches, for instance, since it can pay more attention to the object (an issue captured in more explicit detail in the

context of foraging by *Lloyd and Dayan, 2018*). This underpins the transition between the two types of approach for non-timid animals. In simulations, when $G(t)$ is high, the agent has a high motivation to explore the object, spending only a single turn in the nest state between bouts. In other words, the depletion from $G_0$ substantially influences the time point at which approach makes a transition from peak to steady state; the steady-state time then depends on the dynamics of depletion (when at the object) and replenishment (when at the nest). In particular, in the steady-state phases, the agent must wait multiple turns at the nest for $G(t)$ to regenerate so that informational reward once again exceeds the potential cost of hazard.

Finally, the animal is also motivated to approach by instrumental informational reward arising from the hazard function (which can be exploited to collect more future reward) – according to a standard Bayes-adaptive bonus mechanism (*Duff, 2002b*).

## Conditional value at risk sensitivity

Along with varying degrees of pessimism in their prior over the hazard function, the mice could have different degrees of risk sensitivity in the aspect of the return that they seek to optimize. There are various ways in which the mice might be risk-sensitive. Following *Gagne and Dayan, 2022*, we consider a form called nested conditional value at risk (nCVaR). In general, CVaRα, for risk sensitivity $0 \leq \alpha \leq 1$, measures the expected value in the lower $\alpha$ quantile of returns – thus over-weighting the worse outcomes. The lower $\alpha$, the more extreme the risk aversion; with $\alpha = 1$ being associated with the conventional, risk neutral, expected value of the return. The Bellman updates for BAMDP nCVaR section details the approximate optimization procedure concerned (*Chow et al., 2015*; *Hau et al., 2023*) – it operates by upweighting the probabilities of outcomes with low returns – which come here from detection and expiration. Thus, when $\alpha$ is low, confident and longer bouts are costly, inducing shorter, cautious ones. nCVaRα affects behavior in a similar manner to pessimistic hazard priors, except that nCVaRα acts on both the aleatoric uncertainty of expiring and epistemic uncertainty of detection, while priors only affect the latter. As we will see, despite this difference, we were not able to differentiate pessimistic priors from risk sensitivity using the data in *Akiti et al., 2022*.

## Model fitting

The output of each simulation is a sequence of states which we use to derive summary statistics that can be compared directly with our abstraction of the behavior of a mouse (as in *Figure 1*). This requires us to model transition points in this behavior, and the times involved in each state.

In the model, the transition point from cautious to confident approach happens when the agent first ventures a confident approach; this switch is rarely reversed. Peak to steady-state transition points occur when the model mouse decreases its frequency of bouts, which tends to happen abruptly in the model. We fit the transition points in mouse data by mapping the length of a step in the model to wall-clock time. As in the abstraction of the experimental data, we average the duration (number of turns at the object) and frequency statistics in each phase. We characterize the relative frequencies of the bouts across phase transitions. Frequency mainly governs the total time at or away from the object and is formally defined as the inverse of the number of steps the model spends at the object and the nest.

We use a form of Approximate Bayesian Computation Sequential Monte Carlo (ABCSMC; *Toni et al., 2009*) to fit the elements of our abstraction of the approach behavior of the mice (see Behavior phases and animal groups), namely change points, peak and steady-state durations as well as relative frequencies of bouts. See the Data fitting for details on the fitted statistics. At the core of ABCSMC is the ability to simulate the behavior of model mice for given parameters. We do this by solving the underlying BAMDP problem approximately using receding horizon tree search with a maximum depth of five steps (which covers the longest allowable bout, defined as a subsequence of states where the model mouse goes from the nest to the object and back to the nest).

The full set of parameters includes six for the prior over the hazard function (given that we limit to four the number of time steps the model mouse can stay at the object), the risk sensitivity parameter $\alpha$ for CVaRα, the initial reward pool $G_0$ and the forgetting rate $f$.

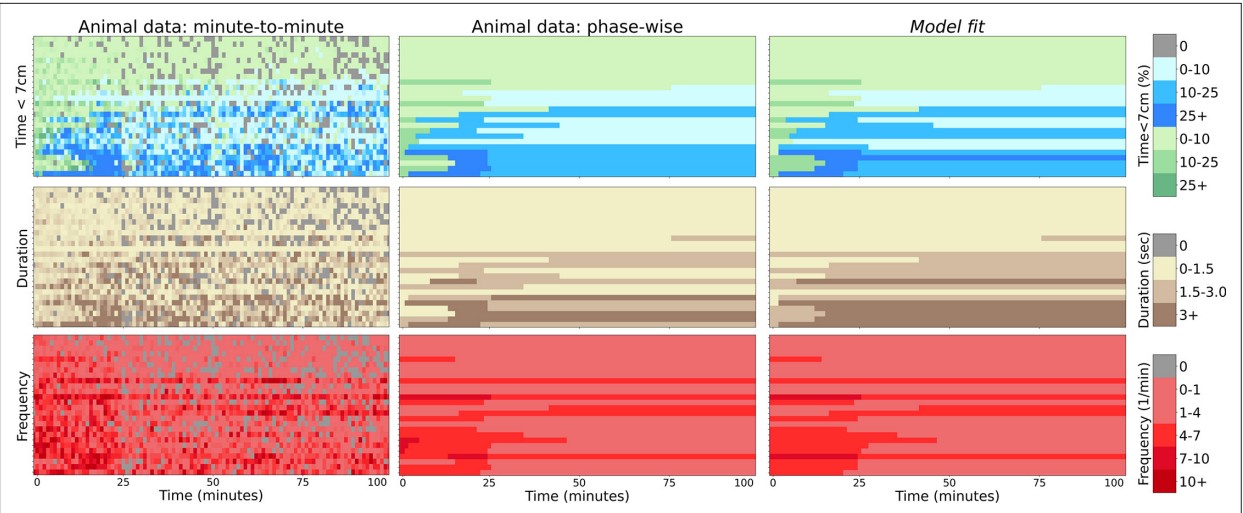

**Figure 5.** Summary of model fit. Left panels: minute-to-minute time the animals spend within 7 cm of the novel object (top), duration (middle), and frequency (bottom). There are 26 animals (one per row) sorted by the group-timidity animal index (see A spectrum of risk-sensitive exploration trajectories). Central panels: the same values averaged over behavioral phases. Right panels: time, duration, and frequency of bouts generated as sample trajectories from the individual fits of the Bayes-adaptive Markov decision process (BAMDP) model. Legend: green/blue distinguishes cautious and confident bouts. The intensity of colors indicates higher values, and gray indicates zeros.

## Explaining exploration schedules with fitted model parameters

### A spectrum of risk-sensitive exploration trajectories

*Figure 5* shows model fits on the 26 mice from *Akiti et al., 2022*. The animal ranking is sorted first by animal group, and second by total time spent near the object. We call this ranking the group-timidity animal index – it slightly differs from the timidity index used in *Akiti et al., 2022* which is only based on total time spent near the object. The model captures many details of the data across the entire spectrum of courage to timidity, explaining the behavior mechanistically. Differing schedules of exploration emerge because of the battle between learning about threat and reward. All animals initially assess risk with a cautious approach, since the costs of potential predation significantly outweigh potential rewards. Brave animals assess risk either with short (length 2 bouts) or medium (length 3 bouts) depending on the hazard priors (*Figure 6a and b versus c and d*). If $E[h_3]$ is high, then the animal performs cautious length 2 bouts, otherwise, it performs cautious length 3 bouts. With more bout experience, the posterior hazard function becomes more optimistic (since there is no actual predator to observe; *Figure 4*), empowering it to take on more risk by staying even longer at the object and performing confident approach. Animals with low $E[h_4]$ perform the longest, confident, length 4 bouts instead of length 3 bouts (*Figure 6a and c versus b and d*). How long brave animals spend assessing risk depends on hazard priors and the risk sensitivity: nCVaR's $\alpha$.

*Figure 7* shows that the fitted hazard priors and risk sensitivity relate to the group-timidity animal index. Brave animals are fitted with higher $\alpha$ and a low slope and high variance (flexibility) hazard prior. In other words, the model brave mouse believes that the hazard probability for long bouts is low in its environment. Timid animals are fitted by lower $\alpha$ and a higher slope, inflexible hazard prior. The parameters for intermediate animals lie between those for brave and timid animals.

$G_0$ determines how much time brave animals spend in the peak-confident exploration phase, or the peak to steady-state change point. Animals with larger $G_0$ tend to have high bout frequencies for a longer period (see *Figure 8*). Finally, how often brave animals revisit the object, which is related to the relative steady-state frequency, is determined by the forgetting rate.

Timid animals have short bouts and continue to assess risk with a cautious approach in the steady state. *Figure 7* shows that their hazard priors are inflexible (low variance), with a high slope, and that they have low $\alpha$. The priors are slow to update, and risk sensitivity causes timid agents to overestimate the probability of bad outcomes, leading to their prolonged cautious behavior. Hence, the reward exploration pool is depleted (i.e. the agent transitions to the steady-state phase) before the agent overcomes its priors. This particular dynamic of approach-drive and hazard function updating leads

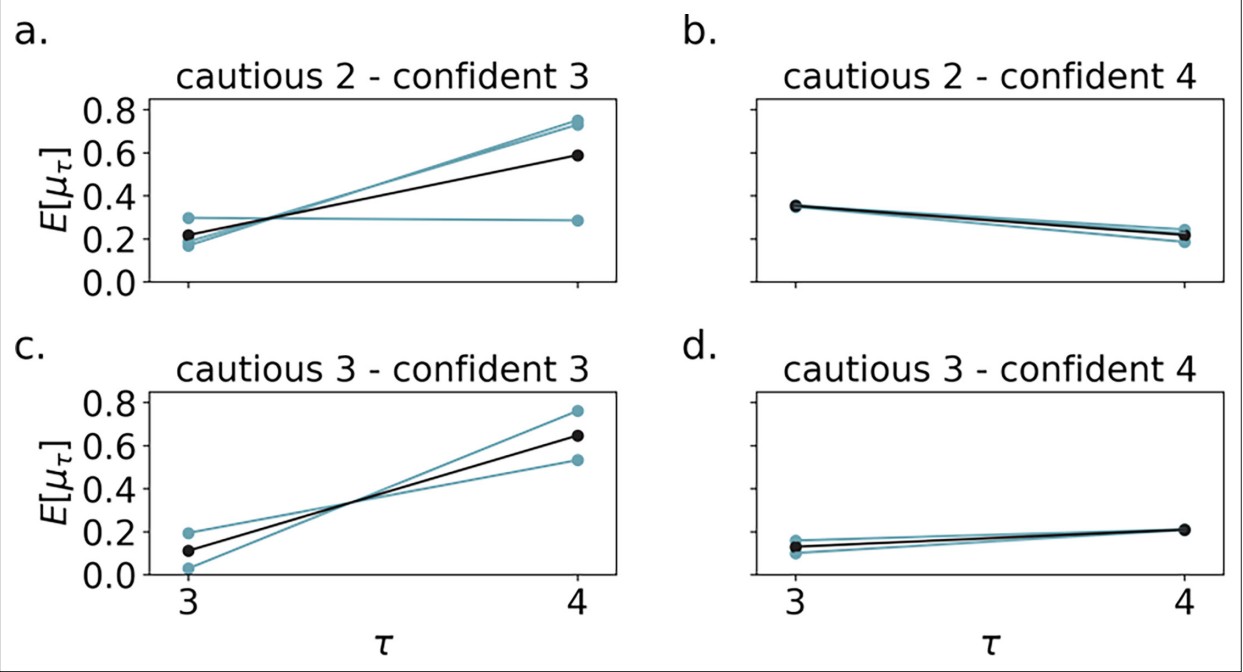

**Figure 6.** The bout durations of brave animals depend on the hazard prior. (**a**) Brave animals that initially perform cautious-2 bouts, then confident-3 bouts. The prior mean $\mu_3$ for $\tau = 3$ is higher than in (**c**) because there is some hazard to overcome before the animal does a duration-3 bout. Blue indicates individual animals and black indicates the mean. The y-axis $E[\mu_\tau]$, shows $\mu_\tau$ averaged over the Approximate Bayesian Computation Sequential Monte Carlo (ABCSMC) posterior particles for each animal. (**b**) Cautious-2 then confident-4 animals. Since the mean $\mu_4$ prior is low, once the animal overcomes the $\tau = 2$ hazard, it quickly transitions from duration 2 to 4. (**c**) Cautious-3, then confident-3 animals. These animals are fitted with a low $\mu_3$ prior and high $\mu_4$ prior because they never perform duration-4 bouts. (**d**) Cautious-3 then confident-4 animals. Since the $\mu_3$ prior is lower than in (**b**), these animals begin with duration-3 bouts.

to self-censoring and neophobia. In the steady-state phase, the agent stays long periods at the nest (how long depends again on the forgetting rate). As a result, the animal (at least during the course of the experiment) never accumulates sufficient evidence to learn the safety of the object or if the object yields rewards. Akiti et al.'s experiment did not last long enough to answer the question of whether all animals, even the most timid ones, eventually perform confident approach. Our model predicts that they will since the agent only accumulates negative evidence for the hazard function. However, with sufficiently low $\alpha$ or pessimistic priors, this may take a very long time.

Intermediate animals, like brave animals, eventually switch to a confident approach to maximize information gained about potential rewards. Similar to brave animals, the cautious to confident transition tends to be later with lower $\alpha$ and steeper, less flexible priors. Intermediate animals perform both cautious and confident bouts with medium duration. This is captured by a hazard prior with smaller $E[h_3]$ and larger $E[h_4]$. The percentage of time spent at the object is relatively constant throughout the experiment for intermediate animals. This can be explained by either large $G_0$ or a high forgetting rate. In other words, the animal is either slow to update its belief about the potential reward at the object, or it expects the reward probability to change quickly.

*Figure 5* also illustrates several limitations of the model. In particular, the duration of bouts can only increase, whereas a few animals exhibit decreasing bout duration between confident-peak and confident-steady-state phases. Furthermore, the model has trouble capturing abrupt changes in duration (from 2 turns to 4) coinciding with an animal's transition from cautious to confident approach.

### Risk sensitivity versus prior belief pessimism

We found that risk sensitivity and prior pessimism could not be teased apart in our model fits. This is illustrated in *Figure 9*. In the ABCSMC posterior distributions, nCVaR's $\alpha$ is correlated with the mean $\mu_2$ for timid and intermediate animals, $\mu_3$ for cautious-2/confident-4 and cautious-2/confident-3 animals, and $\mu_4$ for cautious-2/confident-4 and cautious-3/confident-4 animals. In other words, lower $\alpha$

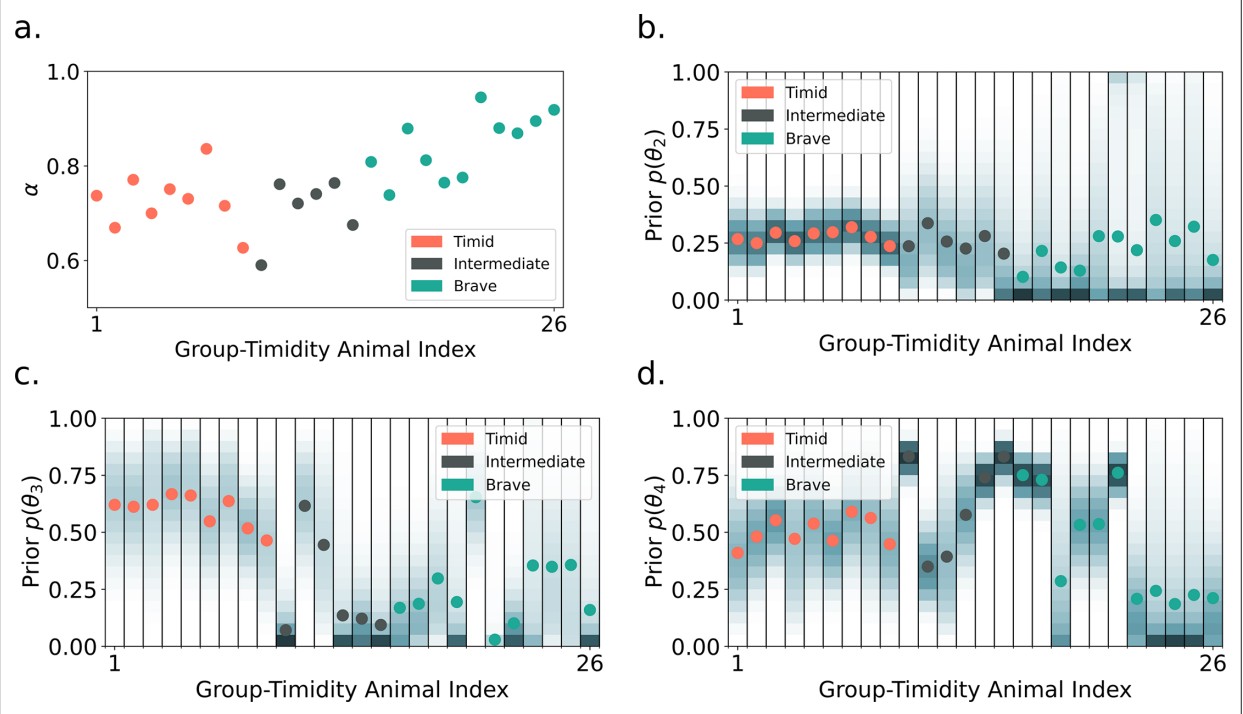

**Figure 7.** Group-level and within-group variation in fitted risk-sensitivity and hazard priors. (**a**) nCVaR's $\alpha$ versus the group-timidity animal index ranking defined in A spectrum of risk-sensitive exploration trajectories. Color indicates the animal group. More timid animals are generally fitted by a lower $\alpha$. Prior hazard parameter for $t = 2$ (**b**), $t = 3$ (**c**), and $t = 4$ (**d**) versus timidity ranking. Dots indicate the mean; the probability density is represented by color where darker means higher density regions. The $t = 2$ prior mean is similar across all animals (timid = $0.28 \pm 0.02$, intermediate = $0.26 \pm 0.04$, brave = $0.22 \pm 0.08$) explaining the short, cautious bouts all animals initially use to assess risk. However, timid animals are best fit with lower variance (inflexible) and higher $t = 3$ and $t = 4$ prior means. This leads to shorter, cautious bouts in the long run. Brave animals are fitted by a low slope (indicated by lower mean for $t = 3$ and $t = 4$) and high variance (flexible) hazard prior. This allows them to perform longer bouts over time. $t = 4$ mean is low (panel d) for brave animals that perform length 4 bouts. Like brave animals, most intermediate animals have flexible, gradual hazards up to $t = 3$.

(higher risk sensitivity) can be traded off against lower (more optimistic) priors to explain the observed risk aversion in animals.

In ablation studies (not shown), we found that it is possible to fit the full range of the behavior equally well with a risk-neutral nCVaR1.0 objective, only varying the hazard priors (with their many extra parameters). There is a technical advantage of fitting both nCVaR$\alpha$ and hazard priors to each animal, namely greater diversity in the particles discovered by ABCSMC.

By contrast, we found that nCVaR$\alpha$ alone, with the same hazard prior for all animals, is incapable of fitting the full range of animal behavior (results not shown). This can be explained by the fact that nCVaR$\alpha$ cannot model the different slopes in the hazard function. For example, a cautious-2/confident-3 animal must be modeled using a high value of $\mu_4$. Starting with the parameters for a cautious-2/confident-4 animal and decreasing $\alpha$ will not create a cautious-2/confident-3 animal. Instead, decreasing $\alpha$ will delay the cautious-to-confident transition of the cautious-2/confident-4 animal and eventually create a cautious-2 timid animal. Therefore, in our task, structured prior beliefs are required to model the detailed behavior of animals. It is not clear in general in which environments one can expect $\alpha$ and priors to be identifiable given the complex interaction of these two sources of risk sensitivity.

## Familiar object novel context

As a contrast to their main experiment, in which mice were exposed to an unfamiliar object in a novel context (UONC), *Akiti et al., 2022* also looked at the consequences of exposing animals to a familiar object in a novel context (FONC), where the animals still habituate in the arena over 2 days but the combination of the object and arena is novel. We fit the behavior of the 9 FONC animals, and, as the closest match, compared this with that of the 11 brave animals in the UONC condition. *Figure 10*

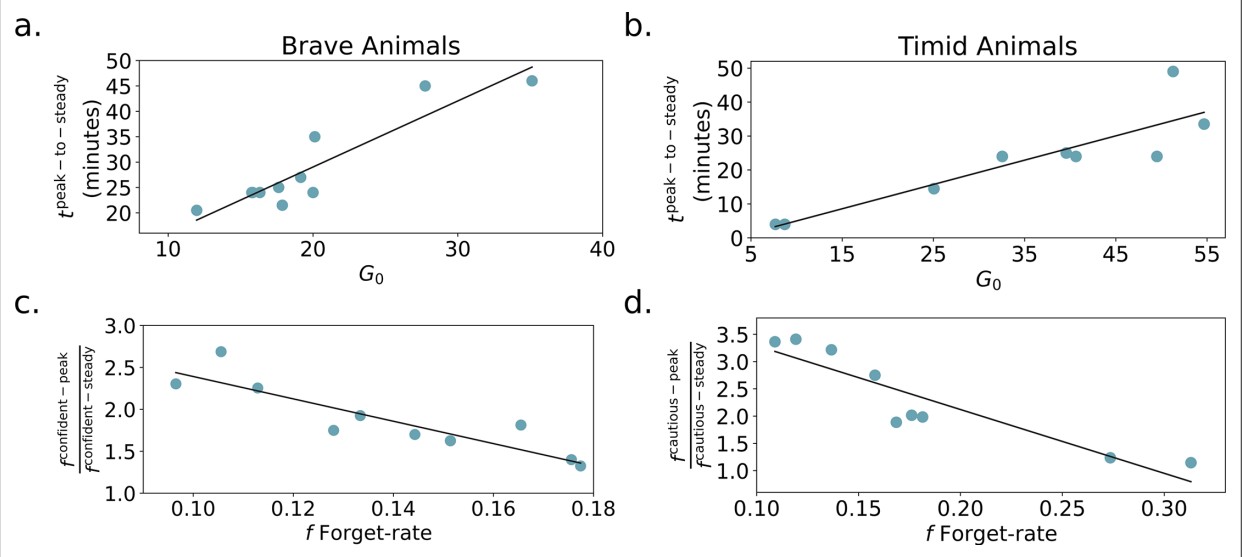

**Figure 8.** Influence of exploration pool and forgetting rate on steady-state behavior. (**a**) The relationship between $G_0$ and the peak to steady-state change point for brave animals. The best fit line is shown in black. Higher $G_0$ means the agent explores longer, hence postponing the change point. (**b**) $G_0$ versus peak to steady-state change point for timid animals. (**c**) Forgetting rate versus steady-state turns at the nest state for brave animals. A higher forgetting rate leads to quicker replenishment of the exploration pool and hence fewer turns at the nest before approaching the object. (**d**) Forgetting rate versus turns at nest timid animal. All correlations are significant with $p < 0.002$.

shows that there are one intermediate and eight brave FONC animals, with the latter having exploration schedules similar to the bravest UONC animals. The eight FONC animals have confident-peak and confident-steady-state phases, meaning their approach decreases in the steady state, suggesting that they are reinvestigating the familiar object for reward.

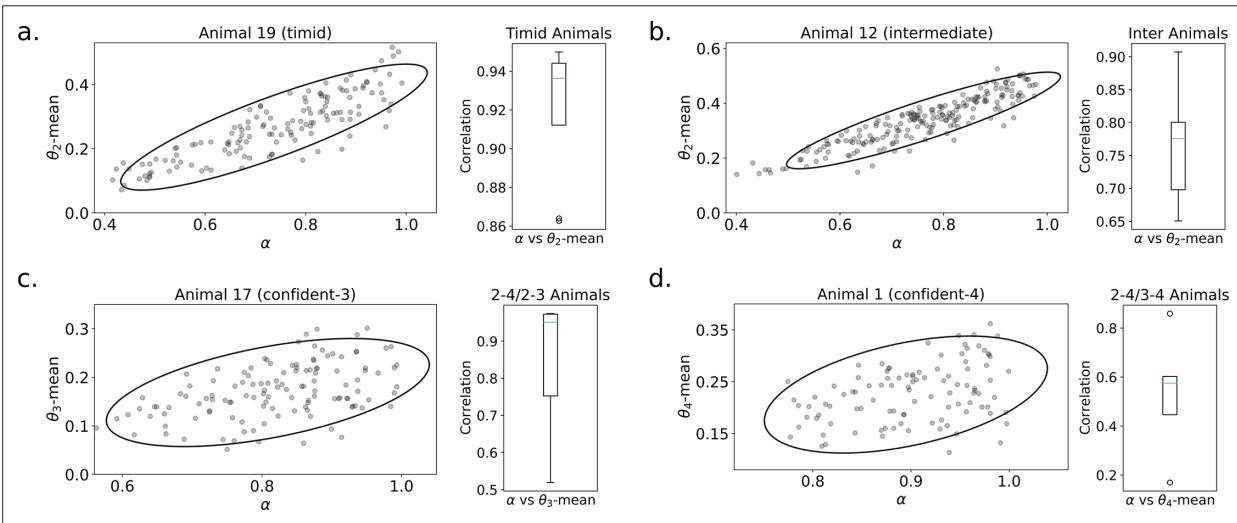

**Figure 9.** Non-identifiability of nCVaR's $\alpha$ against the hazard prior. Animals are labeled using the group-timidity animal index. (**a**) The scatter plot shows the $t = 2$ prior mean ($\mu_2$) versus $\alpha$ for Approximate Bayesian Computation Sequential Monte Carlo (ABCSMC) particles of timid animal 1. The ellipse indicates one standard deviation in a Gaussian density model. Animal 1 (and timid animals generally) can be either fit with a higher $\alpha$ and a higher $\mu_2$, or a lower $\alpha$ and a lower $\mu_2$. The box-and-whisker plot illustrates the correlation between $\mu_2$ and $\alpha$ across all timid animals. (**b**) The scatter plot shows an example intermediate animal 10; the box-and-whisker plot shows $\mu_2$ versus $\alpha$ for the intermediate population. (**c**) The scatter plot shows an example animal 11 from the group containing cautious-2/confident-4 and cautious-2/confident-3 animals. This group of animals starts with duration = 2 bouts and hence must overcome the prior $\mu_3$. The box-and-whisker plot shows $\mu_3$ versus $\alpha$ for the population. (**d**) The scatter plot shows an example animal 25 from the group containing cautious-2/confident-4 and cautious-3/confident-4 animals. This group of animals eventually performs duration = 4 bouts and hence must overcome the prior $\mu_4$. The box-and-whisker plot shows $\mu_4$ versus $\alpha$ for the population. $\alpha$ and $\mu$ are correlated in the ABCSMC posterior for all animals and hence non-identifiable. $p < 0.05$ for all correlations.

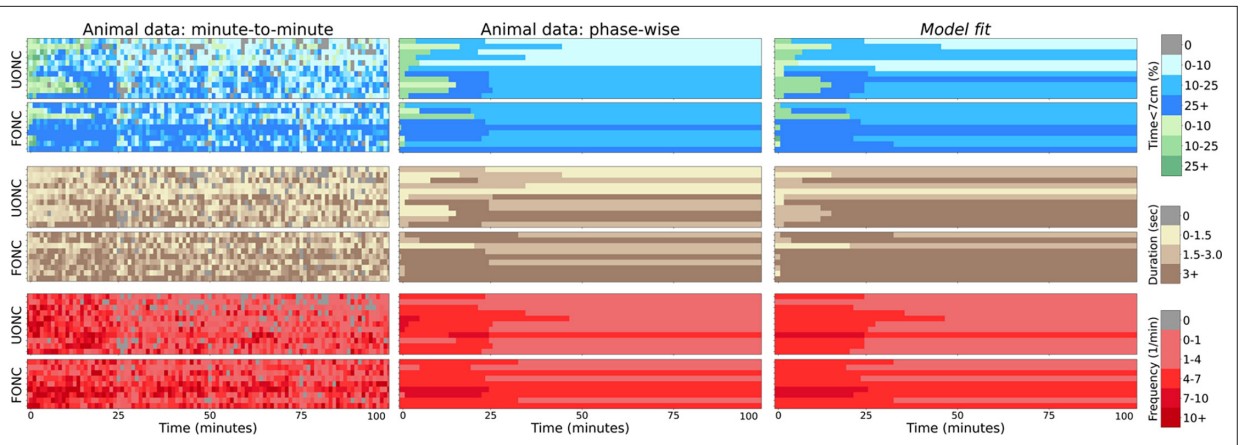

**Figure 10.** Comparing the behavior of FONC and UONC conditions. There are 9 FONC and 11 UONC brave animals (one per row). Left panels: minute-to-minute time the animals spend within 7 cm of the novel object (top), duration (middle), and frequency (bottom). Animals are again sorted by group-timidity animal index but split by experiment condition (UONC then FONC). Central panels: the same values averaged over behavioral phases. Right panels: time, duration, and frequency of bouts generated as sample trajectories from the individual fits of the Bayes-adaptive Markov decision process (BAMDP) model.

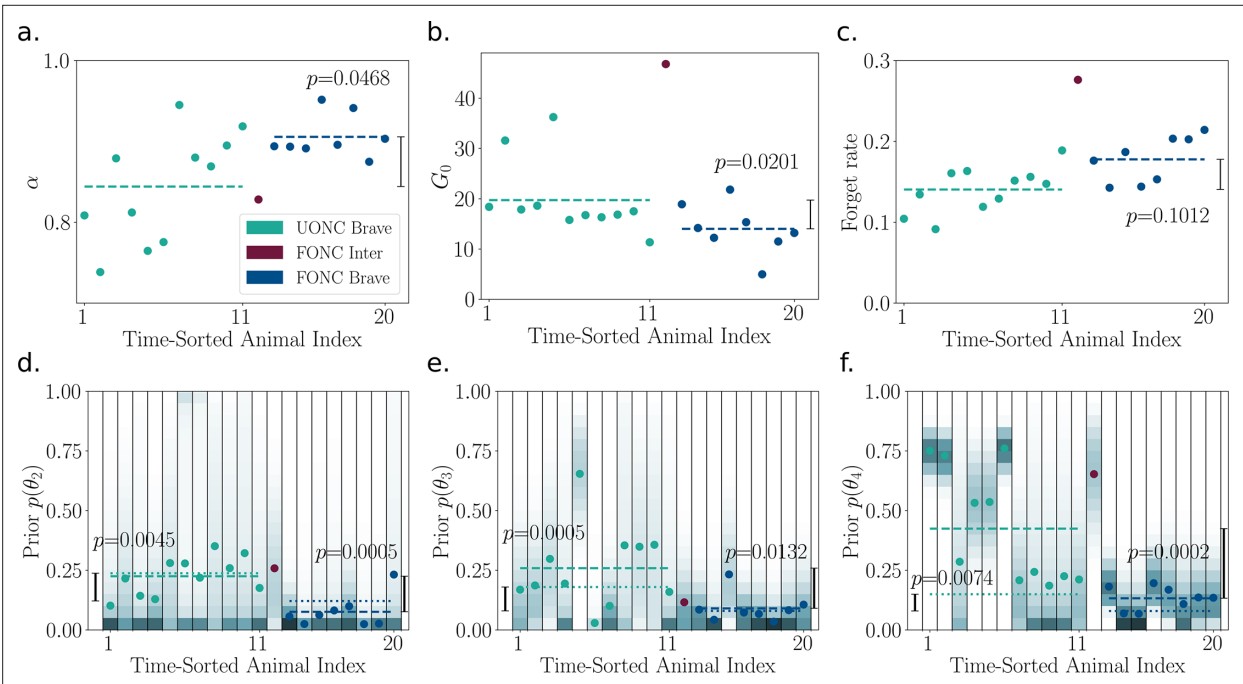

**Figure 11.** Approximate Bayesian Computation Sequential Monte Carlo (ABCSMC) parameter fits of the 9 FONC and 11 UONC animals (with the latter replotted from *Figure 7* for convenience). The x-axis shows group-timidity animal index, but UONC and FONC animals are separated. (**a**) Average nCVaR's $\alpha$ over posterior particles of each animal. Color indicates the animal group. Dashed lines indicate the average (across animals) values of each condition (UONC brave or FONC brave). *p*-values for the Kolmogorov–Smirnov test of condition differences is shown. $p < 0.05$ and therefore the $\alpha$ values of brave FONC animals are significantly higher than those of brave UONC animals. (**b**) Exploration bonus pool, which is also significantly different between FONC and UONC animals. (**c**) Forgetting rate, which is not significantly different between the two conditions. Prior hazard parameter for $t = 2$ (**d**), $t = 3$ (**e**), and $t = 4$ (**f**). The probability density is represented by color where darker means higher density regions. Dots indicate the mean. Dashed lines indicate the average of mean values across animals, while dotted lines indicate the average of standard deviation values across animals. *p*-values testing the difference between the two conditions' means and standard deviations is shown on the right-hand side and left-hand side of the plots, respectively. Brave FONC animals have both significantly lower hazard prior mean and standard deviation than brave UONC animals.

*Figure 11* compares the posteriors of the ABCSMC fit of brave UONC and FONC animals. The *x*-axis shows the group-timidity animal index, but split by experiment condition (UONC then FONC). Compared to brave UONC animals, FONC animals are fitted with higher nCVaR's $\alpha$ and lower hazard priors (average posterior parameters across animals are significantly different according to the Kolmogorov–Smirnov test, $p < 0.05$). Both the hazard prior means and variances are lower for the FONC animals, indicating these animals are more certain of the safety of the object compared to UONC animals. For three animals, the hazard prior means are nearly zero, indicating belief of almost certain safety. This is similar to the hazard function of a brave UONC animal at the end of the experiment. For the other six FONC animals, the hazard prior is high enough to warrant initial cautious bouts suggesting that the novelty of the context has increased their beliefs of the threat level of the familiar object. However, even these animals transition faster to a confident approach than the brave UONC animals. This can be seen in *Figure 10*. *Figure 11b* shows that FONC animals also have on average lower ($p < 0.05$) exploration pool than brave UONC animals. Taken together, these results show that pre-exposure to the object decreases both the animals' beliefs about potential hazards and their motivation to explore the object for reward.

## Discussion

We combined a Bayes-adaptive Markov decision process framework with beliefs about hazards, and a conditional value at risk objective to capture many facets of an abstraction of the substantially different risk-sensitive exploration of individual animals reported by *Akiti et al., 2022*. In the model, behavior reflects a battle between learning about potential threat and potential reward (neither of which actually exists). The substantial individual variability in the schedules of exploratory approach was explained by different risk sensitivities, forgetting rates, exploration bonuses, and prior beliefs about an assumed hazard associated with a novel object. Neophilia arises from a form of optimism in the face of uncertainty, and neophobia from the hazard. Critically, the hazard function is generalizing (reducing the $t = 2$ hazard reduces the $t = 4$ hazard) and monotonic. The former property induces an increasing approach duration over time (*Arsenian, 1943*). Furthermore, the exploration bonus associated with the object regenerates, as if the subjects consider its affordance to be non-stationary (*Dayan et al., 2000*). This encourages even the most timid animals to continue revisiting it. Nevertheless, a main source of persistent timidity is a sort of path-dependent self-censoring (*Dayan et al., 2020*). That is, the agents could be so pessimistic about the object that they never visit it for long enough to overturn their negative beliefs.

In our model, timid behavior can, in principle, arise from either excessive risk sensitivity or overly pessimistic priors about the hazard function. Indeed, CVaR is a probability distortion-based risk measure that is calculated by boosting the probabilities associated with the worst possibilities. Such boosting has obvious parallels with the effect of increasing prior expectations about those worst possibilities – particularly if the consequence of the boosting is a path-dependent reluctance to gather the experience necessary to overwhelm the prior. Given their similar behavioral phenotypes in this task, we duly found that it was not possible to use the model to disentangle the extent to which priors versus values of $\alpha$ were responsible for the observed timidity/bravery. One key difference is that risk aversion continues to affect behavior at the asymptote of learning; something that might be revealed by due choice of a series of environments. Certainly, according to the model, forced exposure (*Huys et al., 2022*) would hasten convergence to the true hazard function and the transition to confident approach.

Due to the complexity of the dataset, we made several rather substantial simplifying assumptions. First, the model employs a particular set of state abstractions, for instance, representing thigmotaxis as a notional 'nest' (*Simon et al., 1994*). Motivated by tail-behind versus tail-exposed in *Akiti et al., 2022*, we model approach using a dichotomy between cautious and confident approach states. This is likely a crude approximation to the continuous and multifaceted nature of animal approach behavior. For example, during approach, animals likely adjust their levels of vigilance continuously (or discretely; *Lloyd and Dayan, 2018*) to monitor threat, and choose different velocities for movement, and different attentional strategies for inspecting the novel object. We hope future works will model these additional behavioral complexities, perhaps with additional internal states, and corroborate these states with neurobiological data.

Second, the model only allows the frequency of approach, and not its duration, to decrease during the steady-state phase – some animals are better fit by decreasing duration. This limitation could be lifted in future models with, for example, a mechanism for boredom causing the animal to retreat when little potential reward remains at the object.

Third, the probability of being detected was the same between cautious and confident approaches, which may not be true in general. Note that the agent decides the type of approach before the bout and is incapable of switching from cautious to confident mid-bout or vice versa.

Fourth, we modeled the relative amount of time the animal spends at the object versus elsewhere in the environment, which depends on the differential risk in the two states. However, it is likely the animals avoid the novel object largely because of the object itself, rather than the potential danger associated with the arena since they spend much less time at the center of the arena during novelty than habituation days.

Finally, we restricted ourselves to a monotonic hazard function for the predator. It would be interesting to experiment with a non-monotonic hazard function instead, as would arise, for instance, if the agent believed that if the predator has not shown up after a long time, then there actually is no predator. Of course, a sophisticated predator would exploit the agent's inductive bias about the hazard function – by waiting until the agent's posterior distribution has settled. In more general terms, the hazard function is a first-order approximation to a complex game-theoretic battle between prey and predator, which could be modeled, for instance, using an interactive IPOMDP (*Gmytrasiewicz and Doshi, 2005*). How the predator's belief about the whereabouts of the prey diminishes could also be modeled game-theoretically, leading to partial hazard resetting rather than the simplified complete resetting in our model.

Our account is model-based, with the mice assumed to be learning the statistics of the environment and engaging in prospective planning (*Mobbs et al., 2020*). By contrast, *Akiti et al., 2022* provide a model-free account of the same data. They suggest that the mice learn the values of threat using an analog of temporal difference learning (*Sutton, 1988*) and explain individual variability as differences in value initialization (*Akiti et al., 2022*). The initial values are generalizations from previous experiences with similar objects and are implemented by activity of dopamine in the TS responding to stimuli salience (*Akiti et al., 2022*). By contrast, our model encompasses extra features of behavior such as bout duration, frequency, and type of approach – ultimately arriving at a different mechanistic explanation of neophobia. In the context of our model, TS dopamine could still respond to the physical salience of the novel object but might then affect choices by determining the potential cost of the encountered threat (a parameter we did not explore here) or perhaps the prior on the hazard function. An analogous mechanism may set the exploration pool or the prior belief about reward – perhaps involving projections from other dopamine neurons, which have been implicated in novelty in the context of exploration bonuses (*Kakade and Dayan, 2002*) and information-seeking for reward (*Ogasawara et al., 2022*; *Bromberg-Martin and Hikosaka, 2009*).

CVaR is known to come in different flavors in the case of temporally extended behavior. *Gagne and Dayan, 2021* introduce two alternative time-consistent formulations of CVaR: nested CVaR (nCVaR) and precommitted CVaR (pCVaR). nCVaR and pCVaR both enjoy Bellman equations which make it possible to compute approximately optimal policies without directly computing whole distributions of the outcomes. We use nCVaR in this study for its computational efficiency. There is, of course, great current interest in distributional reinforcement learning, which does acquire such whole distributions, not the least because of prominent observations linking non-linearities in the response functions of dopamine neurons to methods for learning distributions of outcomes (*Dabney et al., 2020*; *Masset et al., 2023*; *Sousa et al., 2023*). One functional motivation for considering entire outcome distributions is the possibility of using them to determine risk-sensitive policies (*Gagne and Dayan, 2021*). While it is possible to compute CVaR directly from return distributions, *Gagne and Dayan, 2021* showed that this can lead to temporally inconsistent policies where the agent deviates from its original plans (the authors called this the fixed CVaR or fCVaR measure).

Rather further removed from our model-based methods is work from *Antonov and Dayan, 2023*, who consider a model-free exploration strategy which exploits full return distributions to compute the value of perfect information which is used as a heuristic for trying actions with uncertain consequences. Future works can examine risk-sensitive versions of *Antonov and Dayan, 2023*'s computationally

efficient model-free algorithm as one solution to the burdensome computations in our model-based method.

As reported in *Akiti et al., 2022*, animals in the FONC condition in which the object is familiar (though the context is less so) transition quickly to tail-exposed approach and therefore spend more time near the object compared to animals in the UONC condition. *Akiti et al., 2022* models the FONC animals using low initial mean threat and high initial threat uncertainty. We directly compare the behavior of FONC animals against that of the 11 brave UONC animals, showing that FONC animals make choices that are comparable to the bravest UONC animals. FONC behavior is fit by significantly higher nCVaR's $\alpha$ than brave UONC behavior animals. This is surprising if we interpret $\alpha$ as a trait that is stable through time. Unfortunately, due to the non-identifiability between $\alpha$ and hazard priors, we cannot verify whether $\alpha$ is actually higher for FONC animals than UONC animals. FONC animals are also characterized by both lower hazard prior means and standard deviations compared to UONC animals, implying greater certainty about the object's safety. Furthermore, FONC behavior is fitted with lower exploration pools than brave UONC behavior. Taken together, we can understand the FONC animals as having both lower uncertainty about hazard and reward compared to the brave UONC animals at the start of the experiment. However, the hazard and reward uncertainties are higher than what we might expect of UONC animals at the end of the experiment, suggesting the novel context modulates both of these uncertainties. However, heterogeneity exists between FONC individuals in terms of nCVaR's $\alpha$, hazard priors, and exploration pool, which allows another possibility: that both hazard and reward uncertainty are restored by forgetting during the time that passed between pre-exposure and the experiment.

Our model-based account recovers several behavioral phenotypes in addition to those considered in *Akiti et al., 2022*. First, intermittency in our model emerges from the fact that the (possibly CVaR perturbed) hazard function increases with time spent at the object. Therefore, it is rational for the model mice to retreat to the nest when the probability of detection becomes too high and wait until (they believe) the 'predator has forgotten about them', before venturing to the object again.

Second, we offer an alternative explanation for why animals avoid after risk assessment in a benign environment. In Akiti's model, timid animals perform risk assessment because of the delay in model-free value updating from the initial threat at the object (at timestep $t = 10$ in their account) to the time of decision ($t = 8$). In our model, avoidance arises from a rational trade-off between potential risk and reward: timid animals perform risk assessment because of the potential reward at the object and having found none, cease to approach because, although potential threat is lower than at the outset, it still outweighs the even further-reduced potential reward. The same exhaustion of the exploration bonus explains why the brave animals decrease their approach during the steady state of engagement. If the potential reward is low, there is no reason to return to the object at the initial, high rate of engagement.

Third, the temporally evolving battle between reward and threat also explains why brave animals increase their duration of approach when transitioning from risk assessment to engagement. During a confident approach, the animals harvest the exploration pool faster, at the cost of an increased probability of expiring. For brave animals, the hazard posterior decreases faster than the depletion of the exploration pool, and hence brave animals decide to save on travel costs by exploring the object longer in each bout.

Fourth, timid animals return to the object in the steady state of 'avoidance', albeit at a lower rate than during risk assessment. This was not considered in *Akiti et al., 2022*'s account. In our model, timid animals' steady-state approach is explained by the regenerating exploration pool. Such regeneration is natural if the animals assume that the environment is non-stationary, allowing reward structures to change and thus potentially repaying occasional returns to the object if the potential threat has become sufficiently low. Similarly, the animal may believe that the threat is non-stationary. Threat forgetting may act on longer time scales than reward forgetting in our studied environment and is one possible explanation for the initial non-zero hazard functions of some brave animals in the FONC condition.

Finally, our model shows the multi-faceted nature of timidity during exploration. Not only do animals differ in time spent near the object but also in how quickly they transition from cautious to confident approach and their duration and frequency of approach along their exploration schedules. These proxies for timidity are imperfectly correlated. Indeed, an animal could believe that short bouts ($\tau = 2$) are very safe while long bouts ($\tau = 4$) certainly lead to expiration.

Of course, agents do not need to be fully model-free or model-based. They can truncate model-based planning using model-free values at leaf nodes (*Keramati et al., 2016*). Furthermore, replay-like prioritized model-based updates can update a model-free policy when environmental contingencies change (*Antonov and Dayan, 2023*). Finally, while online BAMDP planning can be computationally expensive, a model-based agent may simply amortize planning into a model-free policy which it can reuse in similar environments or even precompile model-based strategies into an efficient model-free policy using meta-learning (*Wang et al., 2017*). Agents may have faced many different exploration environments with differing reward and threat trade-offs through their lifetimes and even over evolution that they have used to create fast, instinctive model-free policies that resemble prospective, model-based behavior (*Rusu et al., 2016*; *Mattar and Daw, 2018*). In turn, TS dopamine might reflect aspects of MF values or prediction errors that had been trained by an MB system following the precepts we outlined.

In *Akiti et al., 2022*, ablating TS-projecting dopamine neurons made mice 'braver'. They spent more time near the object, performed more tail-exposed approach, and transitioned faster to tail-exposed approach compared to control. In *Menegas et al., 2018*, TS ablation affected the learning dynamics for actual, rather than predicted threat. Both ablated and control animals initially demonstrated retreat responses toward airpuffs, but only control mice maintained this response (*Menegas et al., 2018*). After airpuff punishment, ablated individuals surprisingly did not decrease their choices of water ports associated with airpuffs (while controls did). One possibility is that this additional exposure could have caused acclimatization to the airpuffs in the same way that brave animals in our study acclimatize to the novel object by approaching more, and timid animals fail to acclimatize because of self-censoring. Indeed, future experiments might investigate why punishment avoidance does not occur in ablated animals and whether the same holds in risk-sensitive exploration settings (*Menegas et al., 2018*). In other words, would mice decrease approach after reaching the 'detected' state, as expected by our model, or would they maladaptively continue the same rate of approach? Finally, while our study has focused on threat, *Menegas et al., 2017* showed that TS also responds to novelty and salience in the context of rewards and neutral stimuli. That TS ablated animals spend more, rather than less time near the novel object suggests that the link from novelty to neophilia and exploration bonuses might not be mediated by this structure.

The behavior of the mice in *Akiti et al., 2022* somewhat resembles attachment behavior in toddlers (*Ainsworth, 1964*; *Bowlby, 1955*), albeit with the caregiver's trusty leg (a secure base from which to explore) replaced by thigmotaxis (or, in our case, the notional 'nest'). Characteristic to this behavior is an intermittent exploration strategy, with babies venturing away from the leg for a period before retreating back to its safety. Through the time course of exposure to a novel environment, toddlers progressively venture out longer and farther away, spending more time actively playing with the toys rather than passively observing them in hesitation (*Arsenian, 1943*). This is another example of a dynamic exploratory strategy, putatively arising again from differential updates to beliefs about threats and the rewards in the environment (*Arsenian, 1943*; *Ainsworth, 1964*).

Our data show that there is substantial variation in the degrees of risk sensitivity across the mice. Previous works have reported substantial interpopulation and intrapopulation differences in risk sensitivity in humans which depend on gender, age, socioeconomic status, personality characteristics, wealth, and culture (*Rieger et al., 2015*; *Frey et al., 2017*). Despite the normative appeal of $\alpha = 1$, it is possible that a population may benefit from including individuals with $\alpha$ different from 1.0 or highly negative priors. For example, more cautious individuals could learn from merely observing the risky behavior of less cautious individuals. Furthermore, we have only considered risk sensitivity under epistemic uncertainty in our work. Risk-averse individuals, for instance, with $\alpha < 1$ may be more successful than risk-neutral agents in environments where there are unexpected dangers (unknown unknowns). Risk aversion is thus a temperament of ecological and evolutionary significance (*Réale et al., 2007*).

Consistent with this, variability in timidity during exploration has been reported in other animal species and can be caused by differences in both prior experience and genotype. Fish from predator-dense environments tend to make more inspection approaches but stay further away, avoid dangerous areas (attack-cone avoidance), and approach in larger shoals compared to fish from predator-sparse environments (*Magurran and Seghers, 1990*; *Dugatkin, 1988*; *Magurran, 1986*). *Dugatkin, 1988* and *Magurran, 1986* report significant within-population differences in the inspection behavior of guppies and minnows, respectively. *Brown and Dreier, 2002* directly manipulate the predator

experience of glowlight tetras, leading to changes to inspection behavior. Similar inter- and intra-population differences in timidity have been reported in mammals. In *Coss and Biardi, 1997*, the squirrel population sympatric with the tested predators stayed further away and spent less time facing the predator compared to the allopatric population. Furthermore, the number of inspection bouts differed between litters, between individuals within the same litter, and even between the same individuals at different times during development (*Coss and Biardi, 1997*). In *Kemp and Kaplan, 2011*, marmosets differed in risk aversion when inspecting a potential (taxidermic) predator, but risk aversion was not stable across contexts for some individuals. *FitzGibbon, 1994* reports age differences in inspection behavior – adolescent gazelles inspected cheetahs more than adults or half-growns. Finally, *Mazza et al., 2019*; *Eccard et al., 2020* report substantial individual differences in the foraging behavior of voles in risky environments, and *Lloyd and Dayan, 2018* provide a somewhat general model of foraging under risk.

Inter-individual differences in risk sensitivity are also of critical importance in psychiatry, reflected in a panoply of anxiety disorders (*Butler and Mathews, 1983*; *Giorgetta et al., 2012*; *Maner et al., 2007*; *Charpentier et al., 2017*), along with worry and rumination (*Gagne and Dayan, 2022*). Understanding the spectrum of extreme priors and extreme values of $\alpha$ could have therapeutic implications, adding significance to the search for tasks that can more cleanly separate them.

In conclusion, our model shows that risk-sensitive, normative, reinforcement learning can account for individual variability in exploratory schedules of animals, providing a crisp account of the competition between neophilia and neophobia that characterizes many interactions with an incompletely known world.

## Materials and methods
### BAMDP hyperstate
A BAMDP (*Duff, 2002a*; *Guez et al., 2013*) is an extension of model-based MDP and a special case of a partially observable Markov decision process (POMDP; *Kaelbling et al., 1998*) in which the agent models its uncertainty about the (unchanging) transition dynamics. In a BAMDP, the agent extends its state representation into a hyperstate consisting of the original MDP state $s$, and the belief over the transition dynamics $b(T)$.

In our model $s$ is the conjunction of the 'physical state' (the location of the agent, as shown in *Figure 3*) and the number of turns the agent has spent at the object so far $\tau$. In the general case, $T$ is a $|S| \times |A| \times |S|$ tensor where each element is $p(s, a, s')$ and $|S|$ and $|A|$ are the number of states and actions, respectively. Therefore, $b(T)$ is a probability distribution over (possibly infinite) transition tensors. In our model, all transition probabilities are assumed fixed except for the hazard function probabilities. Therefore, a belief over transition tensors $b(T)$ is a belief over hazard functions $b(h)$. We use a noisy-or hazard function parameterized by a vector of Beta distribution parameters $h(\tau; \vec{\mu}, \vec{\sigma})$, $\vec{\mu} = [\mu_1 \ldots, \mu_\tau]$, $\vec{\sigma} = [\sigma_1 \ldots, \sigma_\tau]$. In totality, the belief over transition tensors $b(T)$ is a belief over parameter vectors $b([\vec{\mu}, \vec{\sigma}])$,

$$b(T) = p(T; \vec{\mu}, \vec{\sigma}) \tag{3}$$

However, to maintain generality in the next section, we derive the Bellman updates using the notation $b(T)$.

Our hyperstate additionally contains the nCVaR static risk preference $\bar{\alpha}$, and the parameters of the heuristic exploration bonus $G, n_0^1, n_0^0$ (see Heuristic exploration bonus pool).

### Bellman updates for BAMDP nCVaR
As for a conventional MDP, the nCVaR objective for a BAMDP can be solved using Bellman updates. We use *Equation 4* which assumes a deterministic, state-dependent, reward,

$$V^*(b(T), s, \bar{\alpha}) = \max_a \left[ r(s) + \gamma \min_{\xi \in \mathcal{U}(\bar{\alpha})} \sum_{s'} \xi(b'(T), s') \bar{T}(s, a, s') V^*(b'(T), s', \bar{\alpha}) \right] \tag{4}$$

$s'$ is the next state and $b'(T)$ is the posterior belief over transition dynamics after observing the transition $(s, a, s')$. $\bar{T}(s, a, s')$ is the expected transition probability,

$$\bar{T}(s,a,s') = \int T(s,a,s')\,b(T)\,dT \tag{5}$$

Proof of *Equation 4*.

$$V^*(b(T),s,\bar{\alpha}) = \max_a\{r(x) + \gamma \min_{\xi\in\mathcal{U}(\bar{\alpha})} \int_{\hat{b}(T),s'}$$
$$\xi(\hat{b}(T),s')\cdot p([b(T),s],a,[\hat{b}(T),s'])\cdot V^*(\hat{b}(T),s',\bar{\alpha})d[\hat{b}(T),s']\}$$

where $\mathcal{U}(\bar{\alpha}) = \{\xi : \xi(\hat{b}(T),s') \in [0,\frac{1}{\alpha}], \int_{\hat{b}(T),s'} \xi(\hat{b}(T),s')p([b(T),s],a,[\hat{b}(T),s']) = 1\}$ is the risk envelope for CVaR (*Chow et al., 2015*). But $p([b(T),s],a,[\hat{b}(T),s'])$ is only non-zero when $\hat{b}(T) = b'(T)$,

$$p([b(T),s],a,[\hat{b}(T),s']) = \int_T p(\hat{b}(T),s' \mid T,s,a)\,b(T)\,dT$$
$$= \int_T \delta(\hat{b}(T) - b'(T))\,T(s' \mid s,a)\,b(T)\,dT$$
$$= \begin{cases} \tilde{T}(s,a,s') = \int_T T(s' \mid s,a)\,b(T)\,dT, & \text{if } \hat{b}(T) = b'(T), \\ 0, & \text{otherwise.} \end{cases}$$

Hence, we can drop the independent integration over $(T)$, and only integrate over $s'$,

$$V^*(b(T),s,\bar{\alpha}) = \max_a\{r(s) + \gamma \min_{\xi\in\mathcal{U}(\bar{\alpha})} \int_{s'} \xi(b'(T),s')\cdot \bar{T}(s,a,s')\cdot V^*(b'(T),s',\bar{\alpha})ds'\}$$
$$= \max_a\{r(s) + \gamma \min_{\xi\in\mathcal{U}(\bar{\alpha})} \sum_{s'} \xi(b'(T),s')\cdot \bar{T}(s,a,s')\cdot V^*(b'(T),s',\bar{\alpha})\} \qquad \square$$

Epistemic uncertainty about the transitions only generates risk in as much as it affects the probabilities of realizable transitions in the environment.

## Noisy-or hazard function

$$X_\tau = Z_1 \cup Z_2 \cup \ldots Z_\tau \tag{6}$$

In our model, the hazard function defines a binary detection event $X_\tau$ for each number of turns the agent spends at the object $\tau = 2,3,4$. The predator detects the agent when $X_\tau = 1$. We use a noisy-or hazard function which defines $X_\tau$ as the union of Bernoulli random variables $Z_j \sim \text{Bernoulli}(\theta_j)$ (*Equation 6*) with priors $\theta_j \sim \text{Beta}(\mu_j, \sigma_j)$ for $j = 2,3,4$. *Figure 12* shows the relationships between the random variables in plate notation.

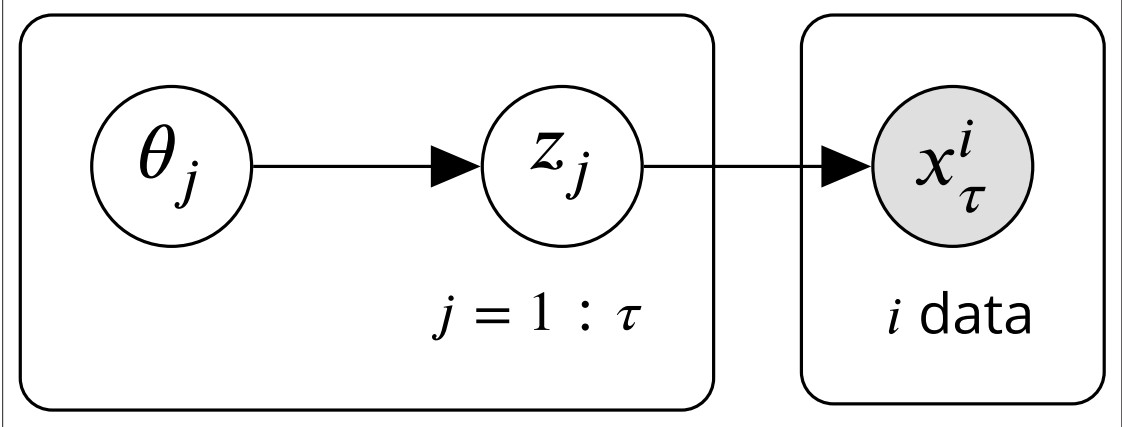

**Figure 12.** Bayes net showing the relationship between the random variables in the noisy-or model. Only $x_\tau$ is shown $x_{\tau+1}$. depends on $z_{t=1:\tau+1}$, and so on.

Posterior inference for the noisy-or model is intractable in the general case (*Jaakkola and Jordan, 1999*). However, there is a closed-form solution for the posterior when the agent only makes negative observations, meaning $x_\tau^i = 0$ (in our case, since there is no actual predator). For example, given a single observation $x_\tau = 0$,

$$
\begin{aligned}
p(\theta_{j=1:\tau}|x_\tau = 0) &= \frac{p(x_\tau = 0|\theta_{j=1:\tau})p(\theta_{j=1:\tau})}{p(x_\tau = 0)} \\
&= \frac{\prod_{j=1:\tau} p(z_j = 0|\theta_t)p(\theta_j)}{p(x_\tau = 0)} \\
&= \frac{\prod_{j=1:\tau}(1 - \theta_j)\mathrm{Beta}(\theta_j; a_j, b_j)}{p(x_\tau = 0)}
\end{aligned}
$$

Here, we switch back to the pseudocount parameterization of the Beta distribution $\mathrm{Beta}(\theta; a, b)$ to exploit its conjugacy,

$$
p(\theta_{j=1:\tau}|x_\tau = 0) \sim \prod_{j=1:\tau} \mathrm{Beta}(\theta_t; a_j, b_j + 1) \tag{7}
$$

Hence, the posterior update simply increments the Beta pseudocounts for the '0' outcomes. The hazard probability is the posterior predictive distribution $h(\tau) = p(x_\tau = 1|D)$ where $D$ are a set of observations of $X_1, X_{2,\tau}$,

$$
p(x_\tau = 1|D) = 1 - \prod_{j=1}^{\tau}(1 - \mu_j) \tag{8}
$$

where $\mu_j = \mathbb{E}[\theta_j]$ is the expected value of the posterior on $\theta_j$.

Proof of *Equation 8*.

$$
\begin{aligned}
p(x_\tau = 1|D) &= 1 - p(x_\tau = 0|D) \\
&= 1 - \int p(x_\tau = 0|\theta_{j=1:\tau})P(\theta_{j=1:\tau}|D)d\theta_{j=1:\tau} \\
&= 1 - \int \prod_{j=1}^{\tau} p(z_j = 0|\theta_j)P(\theta_j|D)d\theta_j \\
&= 1 - \int \prod_{j=1}^{\tau}(1 - \theta_j)\mathrm{Beta}(\theta_j; a_j, \tilde{b}_j)d\theta_j \\
&= 1 - \prod_{j=1}^{\tau} \int (1 - \theta_j)\mathrm{Beta}(\theta_j; a_j, \tilde{b}_j)d\theta_j \\
&= 1 - \prod_{j=1}^{\tau}(1 - \mu_j) \qquad\qquad \square
\end{aligned}
$$

where $\tilde{b}_j$ are the pseudocounts of negative observations after updating the Beta prior with $D$ using *Equation 7*. It can be shown that $h(\tau)$ is recursive,

$$
h(\tau) = h(\tau - 1) + [1 - h(\tau - 1)]\mu_\tau \tag{9}
$$

This recursion has two implications. First, the hazard function is monotonic since $(1 - h(\tau - 1)) > 0$ and $\mu_\tau > 0$. Second, the hazard function generalizes. From *Equation 9* it is clear if $h(\tau - 1)$ increases, then $h(\tau)$ increases. It is this generalization that allows the agent to progressively spend more turns at the object.

## Transforming $\mu, \sigma$ to pseudocount parameterization of Beta distribution

We use the mean $\mu$ and variance $v = \sigma^2$ parameterization of the Beta distribution to get a more uniform sampling of the prior parameter space for ABCSMC fitting. We sample $\mu$ and $\sigma$ from uniform distributions. However, it is more convenient to work with pseudocounts for computing the hazard posterior. Therefore, we transform $\mu$ and $\sigma$ to pseudocounts $a, b$ using the identities below. Note that $v$ must be less than $\mu - \mu^2$ to avoid negative values of $a, b$,

$$
\mu \sim [0, 1] \tag{10}
$$

$$
v \sim [0, \mu - \mu^2] \tag{11}
$$

$$
a = -\frac{\mu(\mu^2 - \mu + v)}{v} \tag{12}
$$

$$
b = \frac{(\mu - 1)(\mu^2 - \mu + v)}{v} \tag{13}
$$

## Heuristic exploration bonus pool

The heuristic reward function approximates the sort of exploration bonus (***Gittins, 1979***) that would arise from uncertainty about potential exploitable benefits of the object. It incentivizes approach and engagement. In the experiment, there is no actual reward so the motivation is purely intrinsic (***Oudeyer and Kaplan, 2007***). The exploration bonus depletes as the agent learns about the object but regenerates if the agent believes that the object can change over time (or, equivalently, if the agent forgets what it has learned). This regenerating uncertainty can be modeled normatively using POMDPs but is only approximated here. Since we imagine the agent as finding more out about the object through a confident than a cautious approach, the former generates a greater bonus per step, but also depletes it more quickly.

We model the exploration-based reward as an exponentially decreasing resource. $G(t)$ is the 'exploration bonus pool' and can be interpreted as the agent's remaining motivation to explore in the future. We fit the size of the initial exploration pool $G(0) = G_0$ to the behavior of each animal. During planning, the agent imagines receiving rewards at the cautious and confident object states proportional to $G(t)$,

$$\hat{r}_{\text{cautious}} = \omega_{\text{cautious}} \cdot G(t) \tag{14}$$

$$\hat{r}_{\text{confident}} = \omega_{\text{confident}} \cdot G(t) \tag{15}$$

$$\hat{r}_{\text{cautious}} < \hat{r}_{\text{confident}} \tag{16}$$

On every turn at the cautious or confident object states, the agent extracts reward $_{\text{cautious}}$ or $_{\text{confident}}$ from its budget $G$, depleting $G$ at rates $\omega_{\text{cautious}}$ or $\omega_{\text{confident}}$. This leads to an exponential decrease in $G(t)$ with turns spent at the object, which is clear from ***Equation 17***. For example, at the cautious object state, the update to $G(t)$ is,

$$G(t + 1) = G(t) - \hat{r}_{\text{cautious}} = (1 - \omega_{\text{cautious}})G(t) \tag{17}$$

However, a secondary factor affects the update to $G(t)$. $G$ linearly regenerates back to $G_0$ at the forgetting rate $f$ which we also fit for each animal. The full update to the reward pool for spending one turn at the cautious object state is,

$$G(t + 1) = \min\{(1 - \omega_{\text{cautious}})G(t) + f, G_0\} \tag{18}$$

Note that $G(t)$ regenerates by $f$ in all states, not only at the object states. We use linear forgetting for its simplicity, although other mechanisms such as exponential forgetting are possible.

Finally, for completeness in other environments, the reward the agent imagines receiving also depends on the actual reward it has received in the past. Let $n^1$ and $n^0$ be the number of times the agent has received one or zero reward at the object state, analogous to the pseudocounts of a Beta posterior in a fully Bayesian treatment of reward. Furthermore, let $n_0^1$ and $n_0^0$ be the (fitted) values at $t = 0$. We use $n_0^1 = 1$ and $n_0^0 = 1$. The agent imagines receiving reward

$$r_{\text{cautious}} = \hat{r}_{\text{cautious}} + \frac{n^1}{n^1 + n^0} \tag{19}$$

after spending one turn in the cautious object state. A similar equation applies to the confident object state.

We define the depletion rates as $\omega_{\text{confident}} = \frac{R}{G_0}$ and $\omega_{\text{cautious}} = K \cdot \omega_{\text{confident}}$ with constants $R = 1.1$ and $K = 0.89 < 1.0$. These values were fitted to capture the full range of behavior of the 26 animals.

## Data fitting

Data fitting aims to elucidate individual differences and population patterns in behavior by searching for the model parameters that best describe the behavior of each animal. We map the behavior of model and animals to a shared abstract space using a common set of statistics and then fit the model to data using ABCSMC.

### Animal statistics

To extract animal statistics, we first coarse-grain behavior into phases and subsequently classify the animals into three groups: brave, intermediate, and timid (as described in the main text). This allows

us to maintain the temporal dynamics of the behavior while reducing the dimension of the data. We average the approach type, duration, and frequency over each phase and fit a subset of statistics that capture the high-level temporal dynamics of behavior of animals in each group.

The behavior of brave animals comes in three phases: cautious, confident-peak, and confident-steady state. We fit five statistics: the transition time from cautious to confident-peak phase $t^{\text{cautious-to-confident}}$, the transition time from confident-peak to confident-steady-state phase $t^{\text{peak-to-steady}}$, the average durations during the cautious and confident-peak phases $d^{\text{cautious}}, d^{\text{peak-confident}}$, and the ratio of confident-peak and confident-steady-state phases' frequencies $\frac{f^{\text{confident-peak}}}{f^{\text{confident-steady}}}$.

Intermediate animals only exhibit two phases: cautious and confident. We fit four statistics: the transition time from cautious to confident phase $t^{\text{cautious-to-confident}}$, the durations of the two phases $d^{\text{cautious}}, d^{\text{confident}}$, and the ratio of the cautious and confident phases frequencies $\frac{f^{\text{cautious}}}{f^{\text{confident}}}$. However, one limitation of the model is that frequency can only decrease, not increase, because of the dynamics of depletion and replenishment of the exploration bonus pool. Hence, we instead fit $\max\{\frac{f^{\text{cautious}}}{f^{\text{confident}}}, 1.0\}$.

Timid animals also only exhibit two phases, albeit different ones from the intermediate animals: cautious-peak and cautious-steady-state. We fit four statistics: the transition time from cautious-peak to cautious-steady-state phase $t^{\text{peak-to-steady}}$, the durations of the two phases $d^{\text{cautious-peak}}$ and $d^{\text{cautious-steady}}$, and the ratio of the frequencies of the two phases $\frac{f^{\text{cautious-peak}}}{f^{\text{cautious-steady}}}$.

## Model statistics

By design, our BAMDP agent also enjoys a notion of bouts and behavioral phases. We map the behavior of the agent to the same abstract space of duration, frequency, and transition time statistics as the animals to allow the fitting.

We consider the agent as performing a bout when it leaves the nest, stays at the object state for some turns, and finally returns to the nest. We parse bouts and behavioral phases from the overall state trajectory of the agent which, like the animals, has what we can describe as contiguous periods of cautious or confident approach and low or high approach frequency.

The transition from the cautious to the confident phase (measured in the number of turns) is when the model begins visiting the confident-object state rather than the cautious-object state (this transition never happens for low $\bar{\alpha}$). The transition from peak to steady-state phase is when the model starts spending >1 consecutive turns at the nest (to regenerate $G$), which happens when $G$ reaches its steady-state value determined by the forgetting rate. We linearly map the agent's transition times (in units of turns) to the space of animals' transition times (units of minutes) using the relationship: 2 turns to 1 min. Therefore, the agent is simulated for 200 turns corresponding to 100 min in the experiment.

Bout duration is naturally defined as the number of consecutive turns the agent spends at the object. Because the agent lives in discrete time, we map its duration (units of turns) to the space of animal duration (units of seconds) using the formula,

$$d_{\text{animal}} = 0.75 + 1.5(d_{\text{agent}} - 2) \tag{20}$$

Hence, the agent is capable of having durations from 0.75 to 3.75 s. This captures a large range of the animals' phase-averaged durations.

We define the momentary frequency with which the agent visits the object as the inverse of the period, which is the number of turns between two consecutive bouts (sum of turns at nest and object

**Table 1.** Table of Approximate Bayesian Computation Sequential Monte Carlo (ABCSMC) parameters.

| Parameter | Description | Value |
|---|---|---|
| $T$ | Number of populations | 30 |
| $B$ | Population size | 100 |
| $\epsilon_t$ | Set adaptively to lowest 30-percentile | |
| $\pi(\theta)$ | Prior distributions for fitted parameters | Uniform |
| $K_t(\theta \mid \theta^*)$ | Transition kernel | $\mathcal{N}(0, \Sigma)$ |
| $d(x, x_0)$ | Distance function | $L_1$ distance |

states). Frequency ratios are computed by dividing the average periods of two phases (in units of turns) and are unitless. Hence, no mapping between agent and animal frequency ratios is necessary.

## Approximate Bayesian computation

We fit each of the 26 animals from *Akiti et al., 2022* separately using an ABCSMC algorithm (*Toni et al., 2009*). We use an adaptive acceptance threshold schedule that sets $\epsilon_t$ to the lowest 30-percentile of distances $d(x, x_0)$ in the previous population. We use a Gaussian transition kernel $K_t(\theta|\theta^*) = \mathcal{N}(0, \Sigma)$, where the bandwidth of $\Sigma$ is set using the Silverman heuristic. We ran ABCSMC for $T = 30$ populations for each animal, but most animals converged earlier. We used uniform priors. *Table 1* contains a list of ABCSMC parameters.

Given agent statistics $\mathbf{x}$ and animal statistics $\mathbf{x}_0$ in a joint space, we compute the ABC distance $d(\mathbf{x}, \mathbf{x}_0)$ using the a normalized $L_1$ distance function.

$$d(\mathbf{x}, \mathbf{x}_0) = \frac{1}{n} \sum_{i=1}^{n} \frac{1}{C^i(x^i)} \left| x^i - x_0^i \right| \tag{21}$$

where $i$ indexes the statistics. $C^i(x^i)$ is a normalization constant that depends on the statistic and possibly the value $x^i$. Normalization is necessary because the statistics have different units and value ranges.

We normalize durations using a constant $C^i(x^i) = 4.0$ s. We normalize the transition times using a piece-wise linear function to prevent extremely small or large values from dominating the distance,

$$C^i(x^i) = \min(30, 10 + 0.8 \max(x^i - 5, 0)) \tag{22}$$

We also normalize the frequency ratio using a piece-wise linear function,

$$C^i(x^i) = \min(20, 2 + \frac{18}{19} \max(x^i - 1, 0)) \tag{23}$$

## Acknowledgements

We are grateful to Chris Gagne, Vikki Neville, Mike Mendl, Elizabeth S Paul, Richard Gao, and particularly Mitsuko Watabe-Uchida for their helpful discussion and feedback. Funding was from the Max Planck Society and the Humboldt Foundation. Open access funding provided by Max Planck Society. PD is a member of the Machine Learning Cluster of Excellence, EXC number 2064/1 – Project number 39072764 and of the Else Kröner Medical Scientist Kolleg 'ClinbrAIn: Artificial Intelligence for Clinical Brain Research'. We thank the IT team from the Max Planck Institute for Biological Cybernetics for technical support.

## Additional information

### Funding

| Funder | Grant reference number | Author |
|---|---|---|
| Max Planck Society | | Peter Dayan |
| Alexander von Humboldt Foundation | | Peter Dayan |
| Deutsche Forschungsgemeinschaft | Machine Learning Cluster of Excellence, EXC number 2064/1 – Project number 39072764 | Peter Dayan |
| Else Kröner-Fresenius-Stiftung | ClinbrAIn: Artificial Intelligence for Clinical Brain Research | Peter Dayan |

| Funder | Grant reference number | Author |
| --- | --- | --- |

The funders had no role in study design, data collection, and interpretation, or the decision to submit the work for publication. Open access funding provided by Max Planck Society.

## Author contributions

Tingke Shen, Conceptualization, Data curation, Software, Formal analysis, Investigation, Visualization, Methodology, Writing – original draft, Writing – review and editing; Peter Dayan, Conceptualization, Supervision, Funding acquisition, Investigation, Methodology, Writing – original draft, Project administration, Writing – review and editing

## Author ORCIDs

Tingke Shen ⓘ https://orcid.org/0009-0000-0286-5663
Peter Dayan ⓘ https://orcid.org/0000-0003-3476-1839

Reviewer #1 (Public review): https://doi.org/10.7554/eLife.100366.3.sa1
Reviewer #3 (Public review): https://doi.org/10.7554/eLife.100366.3.sa2
Author response https://doi.org/10.7554/eLife.100366.3.sa3

# Additional files

## Supplementary files

MDAR checklist

## Data availability

The analysis code used in this study is publicly available at GitHub (copy archived at *Shen, 2025*).

The following previously published dataset was used:

| Author(s) | Year | Dataset title | Dataset URL | Database and Identifier |
| --- | --- | --- | --- | --- |
| Watabe-Uchida M, Akiti K, Tsutsui-Kimura I, Xie Y, Mathis A, Markowitz J, Anyoha R, Datta S, Weygandt Mathis M, Uchida N | 2023 | Dopamine activity in the tail of the striatum, DeepLabCut and MoSeq during novel object exploration | https://doi.org/10.5061/dryad.41ns1rnh2 | Dryad Digital Repository, 10.5061/dryad.41ns1rnh2 |

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

# Appendix 1

## A recovery analysis

We performed recovery analysis on our ABCSMC fits. The recovery targets were the best-fitting particles for each of the 26 mice. We ran ABCSMC a second time, using the same hyperparameters, to check that we could recover the recovery targets.

*Appendix 1—figure 1* compares the recovery targets against the closest particles in the posterior of the (recovery) ABCSMC fit. Each subplot shows one of the nine fitted parameters: $\mathrm{nCVaR}_\alpha$, $G_0$, the forgetting rate $f$, the three hazard prior means, and the three hazard prior deviations. In general, the ABCSMC fitting algorithm recovers the recovery target reasonably well for all animals, with a minimum $R^2$ value of 0.72.

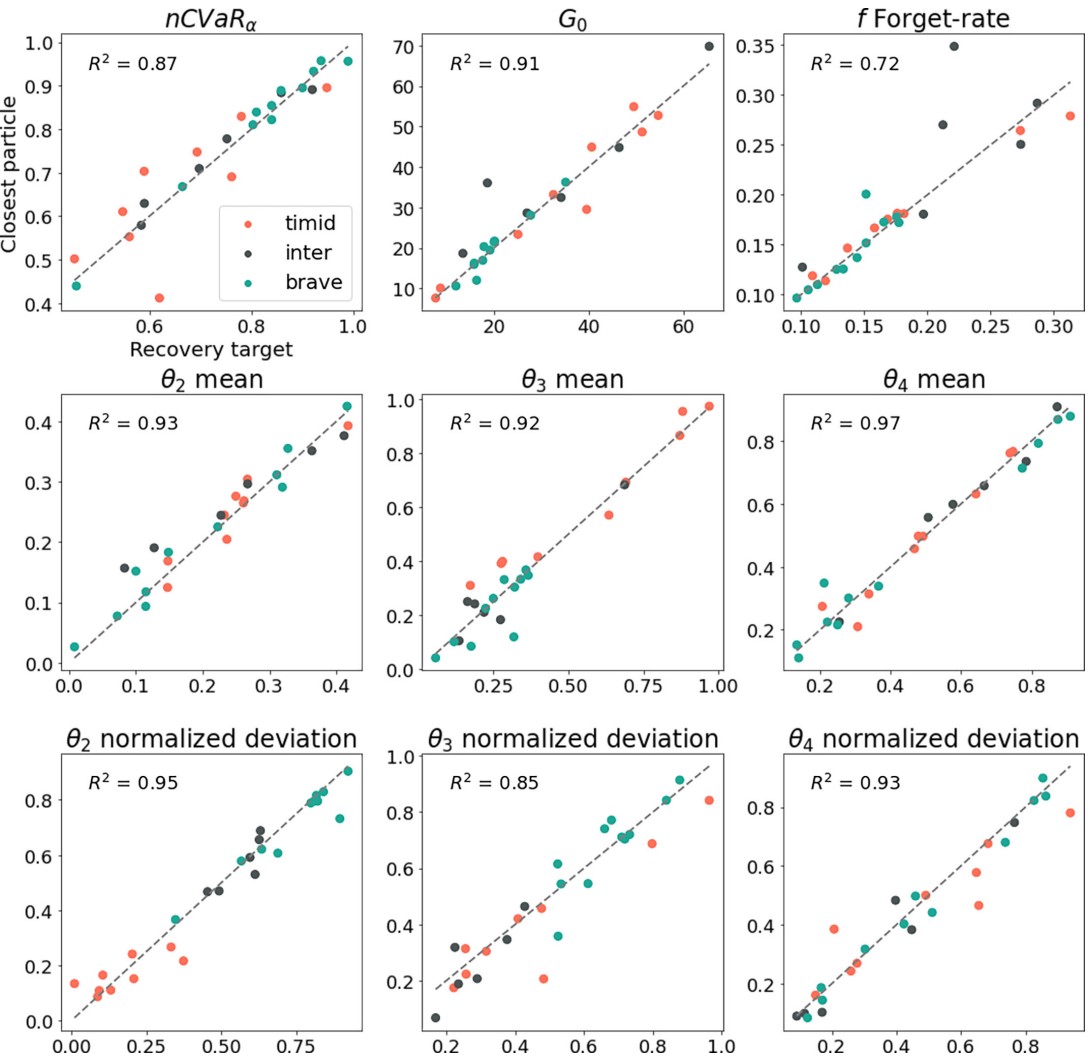

**Appendix 1—figure 1.** Recovery targets versus the closest particles in the ABCSMC posterior. Each subplot plots one of the nine fitted parameters for all 26 animals. The colors of the points indicate the animal group. The gray $y = x$ line represents a perfect recovery of the recovery targets. Most points lie close to the $y = x$ line, suggesting our ABCSMC fitting algorithm has good recoverability.

*Appendix 1—figure 2* compares the recovery targets against the (marginal) means of the ABCSMC posterior. The exploration pool $G_0$ and forgetting rate $f$ are well recovered. However, there is poor recoverability for $\mathrm{nCVaR}_\alpha$ and the prior parameters due to non-identifiability. This is further illustrated in *Appendix 1—figure 3* for a single brave animal. *Appendix 1—figure 3* plots the univariate and bivariate marginals of the ABCSMC posterior. As expected, the recovery targets lie within a narrow range of the posterior distributions for $G_0$ and $f$. For $\mathrm{nCVaR}_\alpha$ and the prior

parameters, the recovery targets are farther from the means of the posterior but still lie within a region of the posterior with support.

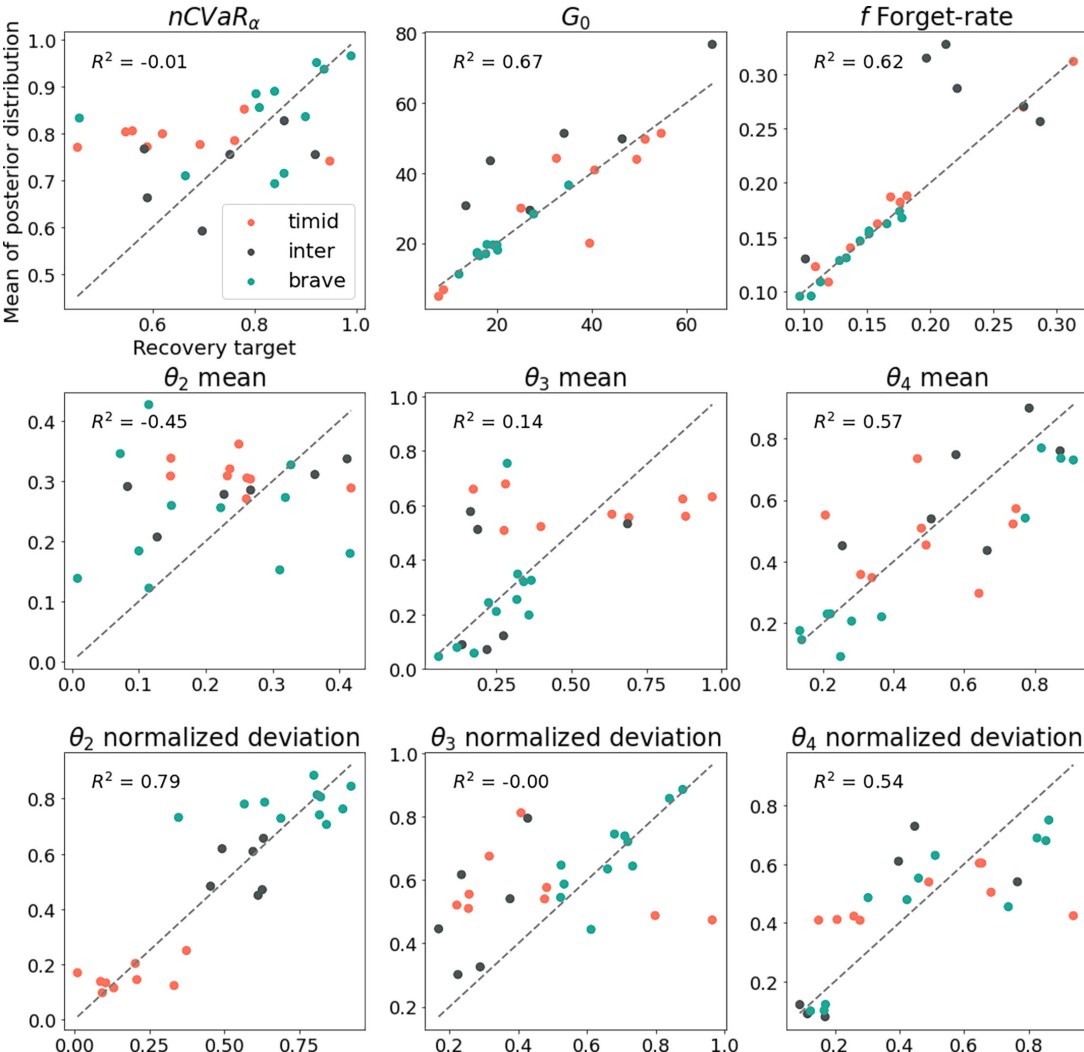

**Appendix 1—figure 2.** Identical to *Appendix 1—figure 1* but the recovery targets are plotted against the (marginal) means of the ABCSMC posterior. We chose the final ABCSMC population for the posterior (population 15). $R^2$ is high for $G_0$ and, $f$ suggesting that these parameters are identifiable. $R^2$ is low for $nCVaR_\alpha$ and the hazard priors due to the non-identifiability discussed in the main text. In particular, $R^2$ is less than 0.0 for $nCVaR_\alpha$ and $\theta_2$-mean suggesting these parameters are the most confounded. However, $R^2$ is high for, $\theta_2$-deviation suggesting $nCVaR_\alpha$ does not confound the flexibility of the hazard function. Finally, the $R^2$ for $\theta_3$ is nearly zero. This is expected because timid and some intermediate animals do not have duration-3 approach, and for these animals, $\theta_3$ can take on arbitrarily large values.

# Marginal and pairwise posterior distributions for animal 24

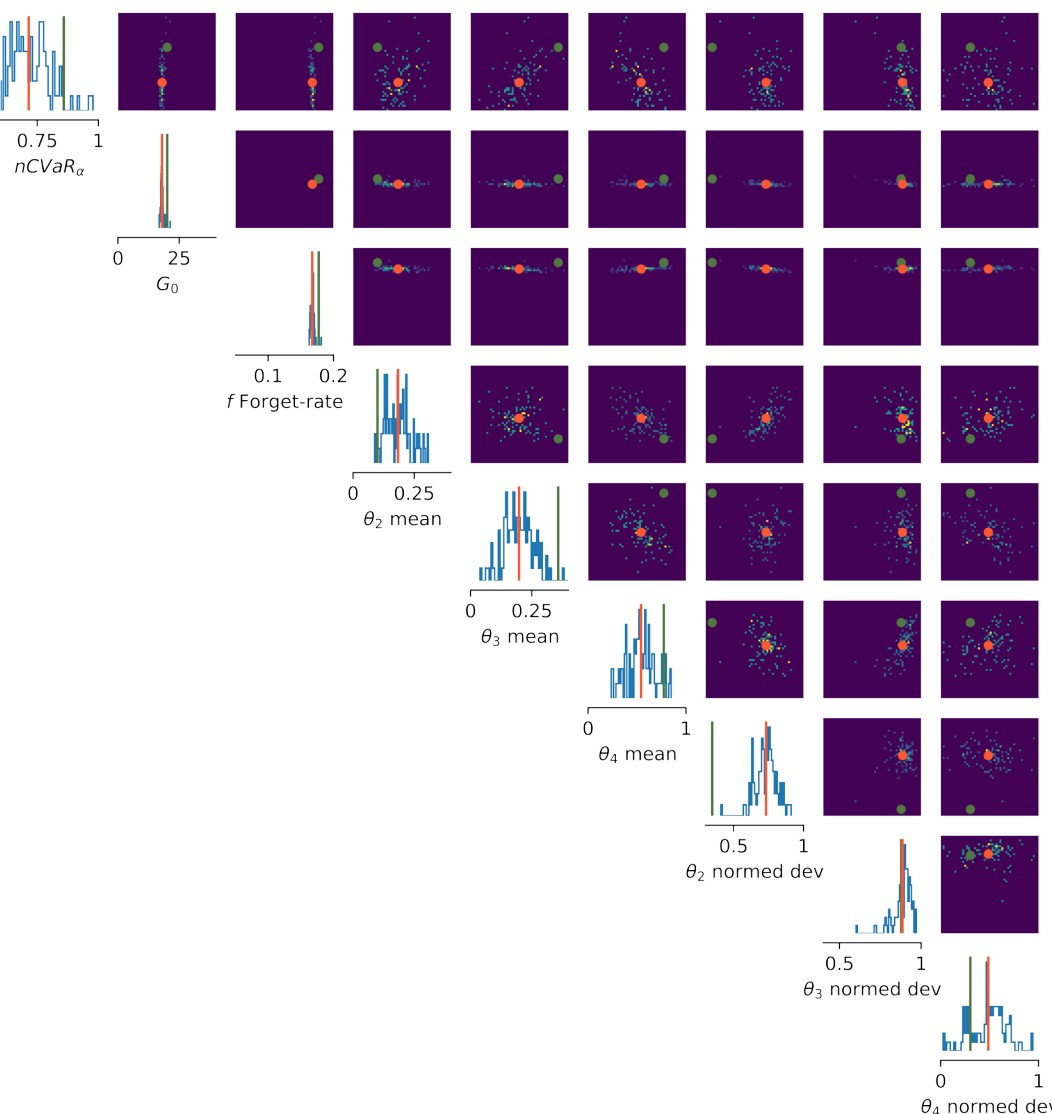

**Appendix 1—figure 3.** The ABCSMC posterior for animal 24. Univariate and bivariate marginals are shown on the diagonal and off-diagonal, respectively. Recovery targets are shown as green vertical lines in univariate plots and green points on bivariate plots. Marginal means are shown in orange. Recovery targets and means are close for $G_0$ and $f$ due to their identifiability. $\mathbf{nCVaR}_\alpha$ and the hazard prior parameters are non-identifiable. Hence, the recovery targets are farther from the mean but still lie in a region of the posterior with support.

