## [Editor Report · eLife Assessment]

Shen et al. present a computational account of individual differences in mouse exploration when faced with a novel object in an open field from a previously published study (Akiti et al.) that relates subject-specific intrinsic exploration and caution about potential hazards to the spectrum of behaviors observed in this setting. Overall, this computational study is an **important** contribution that leverages a very general modeling framework (a Bayes Adaptive Markov Decision Process) to quantify and interrogate distinct drivers of exploratory behavior under potential threat. Given their assumptions, the modeling results are **convincing**: the authors are able to describe a substantial amount of the behavioral features and idiosyncracies in this dataset, and their model affords a normative interpretation related to inherent risk aversion and predation hazard "flexibility" of individual animals and should be of broad interest to researchers working to understand open-ended exploratory behaviors.

---

## [Referee Report · Reviewer #1 (Public review)]

Summary:

This work computationally characterized the threat-reward learning behavior of mice in a recent study (Akiti et al.), which had prominent individual differences. The authors constructed a Bayes-adaptive Markov decision process model, and fitted the behavioral data by the model. The model assumed (i) hazard function staring from a prior (with free mean and SD parameters) and updated in a Bayesian manner through experience (actually no real threat or reward was given in the experiment), (ii) risk-sensitive evaluation of future outcomes (calculating lower 𝛼 quantile of outcomes with free 𝛼 parameter), and (iii) heuristic exploration bonus. The authors found that (i) brave animals had more widespread hazard priors than timid animals and thereby quickly learned that there was in fact little real threat, (ii) brave animals may also be less risk-aversive than timid animals in future outcome evaluation, and (iii) the exploration bonus could explain the observed behavioral features, including the transition of behavior from the peak to steady-state frequency of bout. Overall, this work is a novel interesting analysis of threat-reward learning, and provides useful insights for future experimental and theoretical work. However, there are several issues that I think need to be addressed.

Strengths:

- This work provides a normative Bayesian account for individual differences in braveness/timidity in reward-threat learning behavior, which complements the analysis by Akiti et al. based on model-free threat reinforcement learning.

- Specifically, the individual differences were characterized by (i) the difference in the variance of hazard prior and potentially also (ii) the difference in the risk-sensitivity in evaluation of future returns.

Weakness:

- Theoretically the effect of prior is diluted over experience whereas the effect of biased (risk-aversive) evaluation persists, but these two effects could not be teased apart in the fitting analysis of the current data.

- It is currently unclear how (whether) the proposed model corresponds to neurobiological (rather than behavioral) findings, different from the analysis by Akiti et al.

Comments on revisions:

The authors have adequately replied to all the concerns that I raised in my review of the original manuscript. I do not have any remaining concern, and I am now more convinced that this work provides novel important insights and stimulates future experimental and theoretical examinations.

---

## [Referee Report · Reviewer #3 (Public review)]

Summary:

The manuscript presents computational modelling of the behaviour of mice during encounters with novel and familiar objects, originally reported in Akiti et al. (Neuron 110, 2022). Mice typically perform short bouts of approach followed by retreat to a safe distance, presumably to balance exploration to discover possible reward with the potential risk of predation. However, there is considerable heterogeneity in this exploratory behaviour, both across time as an individual subject becomes more confident in approaching the object, and across subjects; with some mice rapidly becoming confident to closely explore the object, while other timid mice never become fully confident that the object is safe. The current work aims to explain both the dynamics of adaptation of individual animals over time, and the quantitative and qualitative differences in behaviour between subjects, by modelling their behaviour as arising from model-based planning in a Bayes adaptive Markov Decision Process (BAMDP) framework, in which the subjects maintain and update probabilistic estimates of the uncertain hazard presented by the object, and rationally balance the potential reward from exploring the object with the potential risk of predation it presents.

In order to fit these complex models to the behaviour the authors necessarily make substantial simplifying assumptions, including coarse-graining the exploratory behaviour into phases quantified by a set of summary statistics related to the approach bouts of the animal. Inter-individual variation between subjects is modelled both by differences in their prior beliefs about the possible hazard presented by the object, and by differences in their risk preference, modelled using a conditional value at risk (CVaR) objective, which focuses the subject's evaluation on different quantiles of the expected distribution of outcomes. Interestingly, these two conceptually different possible sources of inter-subject variation in brave vs timid exploratory behaviour turn out not to be dissociable in the current dataset as they can largely compensate for each other in their effects on the measured behaviour. Nonetheless, the modelling captures a wide range of quantitative and qualitative differences between subjects in the dynamics of how they explore the object, essentially through differences in how subject's beliefs about the potential risk and reward presented by the object evolve over the course of exploration, and are combined to drive behaviour.

Exploration in the face of risk is a ubiquitous feature of the decision-making problem faced by organisms, with strong clinical relevance, yet remains poorly understood and under-studied, making this work a timely and welcome addition to the literature.

Strengths:

- Individual differences in exploratory behaviour are an interesting, important, and under-studied topic.

- Application of cutting-edge modelling methods to a rich behavioural dataset, successfully accounting for diverse qualitative and qualitative features of the data in a normative framework.

- Thoughtful discussion of the results in the context of prior literature.

Limitations:

- The model-fitting approach used of coarse-graining the behaviour into phases and fitting to their summary statistics may not be applicable to exploratory behaviours in more complex environments where coarse-graining is less straightforward.

Comments on revisions:

All recommendations to authors from the first review were addressed in the revised manuscript.

---

## [Author Response]

The following is the authors’ response to the original reviews.

**Reviewer #1 (Public review):**
This work computationally characterized the threat-reward learning behavior of mice in a recent study (Akiti et al.), which had prominent individual differences. The authors constructed a Bayes-adaptive Markov decision process model and fitted the behavioral data by the model. The model assumed (i) hazard function starting from a prior (with free mean and SD parameters) and updated in a Bayesian manner through experience (actually no real threat or reward was given in the experiment), (ii) risk-sensitive evaluation of future outcomes (calculating lower 𝛼 quantile of outcomes with free 𝛼 parameter), and (iii) heuristic exploration bonus. The authors found that (i) brave animals had more widespread hazard priors than timid animals and thereby quickly learned that there was in fact little real threat, (ii) brave animals may also be less risk-aversive than timid animals in future outcome evaluation, and (iii) the exploration bonus could explain the observed behavioral features, including the transition of behavior from the peak to steady-state frequency of bout. Overall, this work is a novel interesting analysis of threat-reward learning, and provides useful insights for future experimental and theoretical work. However, there are several issues that I think need to be addressed.Strengths:(1) This work provides a normative Bayesian account for individual differences in braveness/timidity in reward-threat learning behavior, which complements the analysis by Akiti et al. based on model-free threat reinforcement learning.(2) Specifically, the individual differences were characterized by (i) the difference in the variance of hazard prior and potentially also (ii) the difference in the risk-sensitivity in the evaluation of future returns.Weakness:(1) Theoretically the effect of prior is diluted over experience whereas the effect of biased (risk-aversive) evaluation persists, but these two effects could not be teased apart in the fitting analysis of the current data.(2) It is currently unclear how (whether) the proposed model corresponds to neurobiological (rather than behavioral) findings, different from the analysis by Akiti et al.

We thank reviewer #1 for their useful feedback which we’ve used to improve the discussion, formatting and clarity of the paper, and for highlighting important questions for future extensions of our work.

Major points:(1) Line 219It was assumed that the exploration bonus was replenished at a steady rate when the animal was at the nest. An alternative way would be assuming that the exploration bonus slowly degraded over time or experience, and if doing so, there appears to be a possibility that the transition of the bout rate from peak to steady-state could be at least partially explained by such a decrease in the exploration bonus.

Section 2.2.3 explains the mechanism of the exploration bonus which motivates approach. We think that the mechanism suggested by the reviewer is, in essence, what is happening in the model. The exploration pool is indeed depleted over time or bouts of experience at the object. In the peak confident phase for brave animals and the peak cautious phase for timid animals, the rate of depletion exceeds the rate of regeneration, since the agent spends only a single turn at the nest between bouts. In the steady-state phase, the exploration pool has depleted so much previously that the agent must wait multiple turns at the nest for the pool to regenerate to a sufficiently high value to justify approaching the object again.

We have updated section 2.2.3 to explain that agents spend one turn at the nest during peak phase but multiple turns during steady-state phase. Hopefully, this makes our mechanism clear:

“In simulations, when 𝐺(𝑡) is high, the agent has a high motivation to explore the object, spending only a single turn in the nest state between bouts. In other words, the depletion from 𝐺0 substantially influences the time point at which approach makes a transition from peak to steady-state; the steady-state time then depends on the dynamics of depletion (when at the object) and replenishment (when at the nest). In particular, in the steady-state phases, the agent must wait multiple turns at the nest for 𝐺(𝑡) to regenerate so that informational reward once again exceeds the potential cost of hazard.“

(2) Line 237- (Section 2.2.6, 2.2.7, Figures 7, 9)I was confused by the descriptions about nCVaR. I looked at the cited original literature Gagne & Dayan 2022, and understood that nCVaR is a risk-sensitive version of expected future returns (equation 4) with parameter α (α-bar) (ranging from 0 to 1) representing risk preference. Line 269-271 and Section 4.2 of the present manuscript described (in my understanding) that α was a parameter of the model. Then, isn't it more natural to report estimated values of α, rather than nCVaR, for individual animals in Section 2.2.6, 2.2.7, Figures 7, 9 (even though nCVaR monotonically depends on α)? In Figures 7 and 9, nCVaR appears to be upper-bounded to 1. The upper limit of α is 1 by definition, but I have no idea why nCVaR was also bounded by 1. So I would like to ask the authors to add more detailed explanations on nCVaR. Currently, CVaR is explained in Lines 237-243, but actually, there is no explanation about nCVaR rather than its formal name 'nested conditional value at risk' in Line 237.

Thank you for pointing out this error. We have corrected the paper to use nCVaR to refer to the objective and nCVaR's α, or sometimes just α, to refer to the risk sensitivity parameter and thus the degree of risk sensitivity.

(3) Line 333 (and Abstract)Given that animals' behaviors could be equally well fitted by the model having both nCVaR (free α) and hazard prior and the alternative model having only hazard prior (with α = 1), may it be difficult to confidently claim that brave (/timid) animals had risk-neutral (/risk-aversive) preference in addition to widespread (/low-variance) hazard prior? Then, it might be good to somewhat weaken the corresponding expression in the Abstract (e.g., add 'potentially also' to the result for risk sensitivity) or mention the inseparability of risk sensitivity and prior belief pessimism (e.g., "... although risk sensitivity and prior belief pessimism could not be teased apart").

Thank you for this suggestion, we have duly weakened the wording in the Abstract to say “potentially more risk neutral”:

“Some animals begin with cautious exploration, and quickly transition to confident approach to maximize exploration for reward; we classify them as potentially more risk neutral, and enjoying a flexible hazard prior. By contrast, other animals only ever approach in a cautious manner and display a form of self-censoring; they are characterized by potential risk aversion and high and inflexible hazard priors.”

**Reviewer #2 (Public Review):**
Shen and Dayan build a Bayes adaptive Markov decision process model with three key components: an adaptive hazard function capturing potential predation, an intrinsic reward function providing the urge to explore, and a conditional value at risk (CvaR, closely related to probability distortion explanations of risk traits). The model itself is very interesting and has many strengths including considering different sources of risk preference in generating behavior under uncertainty. I think this model will be useful to consider for those studying approach/avoid behaviors in dynamic contexts.The authors argue that the model explains behavior in a very simple and unconstrained behavioral task in which animals are shown novel objects and retreat from them in various manners (different body postures and patterns of motor chunks/syllables). The model itself does capture lots of the key mouse behavioral variability (at least on average on a mouse-by-mouse basis) which is interesting and potentially useful. However, the variables in the model - and the internal states it implies the mice have during the behavior - are relatively unconstrained given the wide range of explanations one can offer for the mouse behavior in the original study (Akiti et al). This reviewer commends the authors on an original and innovative expansion of existing models of animal behaviour, but recommends that the authors revise their study to reflect the obvious challenges . I would also recommend a reduction in claiming that this exercise gives a normative-like or at least quantitative account of mental disorders.

We thank reviewer #2 for highlighting some of the strengths of our paper as well as pointing out important limitations of Akiti et al’s original study which we’ve inherited as well as some limitations of our own method. We address their concerns below.

We have added a paragraph to the discussion discussing the limitations of the state representation we adopted from Akiti’s study.

(Reviewer #1 had the same concern, see above) “Motivated by tail-behind versus tail-exposed in Akiti et al. (2022), we model approach using a dichotomy between cautious and confident approach states [...]”

We have reduced the suggestion that our model provides an account of mental disorders in the abstract.

Before:

“On the other hand, “timid” animals, characterized by risk aversion and high and inflexible hazard priors, display self-censoring that leads to the sort of asymptotic maladaptive behavior that is often associated with psychiatric illnesses such as anxiety and depression.”

After:

“By contrast, other animals only ever approach in a cautious manner and display a form of self-censoring; they are characterized by potential risk aversion and high and inflexible hazard priors. “

My main comment is that this paper is a very nice model creation that can characterize the heterogeneity rodent behavior in a very simple approach/avoid context (Akiti et al; when a novel object is placed in an arena) that itself can be interpreted in a multitude of ways. The use of terms like "exploration", "brave", etc in this context is tricky because the task does not allow the original authors (Akiti et al) to quantify these "internal states" or "traits" with the appropriate level of quantitative detail to say whether this model is correct or not in capturing the internal states that result in the rodent behavior. That said, the original behavioral setup is so simple that one could imagine capturing the behavioral variability in multiple ways (potentially without evoking complex computations that the original authors never showed the mouse brain performs). I would recommend reframing the paper as a new model that proposes a set of internal states that could give rise to the behavioral heterogeneity observed in Akiti et al, but nonetheless is at this time only a hypothesis. Furthermore, an explanation of what would be really required to test this would be appreciated to make the point clearer.

We thought very hard about using terms that might be considered to be anthropomorphic such as ‘timid’ and ‘brave’. We are, of course, aware, of the concerns articulated by investigators such as LeDoux about this. However, we think that, provided that we are clear on the first appearance (using ‘scare’ quotes) that we are using them as indeed labels for latent characteristics that capture correlations in various aspects of behaviour, they are more helpful than harmful in making our descriptions understandable.

**Reviewer #3 (Public Review):**
Summary:The manuscript presents computational modelling of the behaviour of mice during encounters with novel and familiar objects, originally reported by Akiti et al. (Neuron 110, 2022) . Mice typically perform short bouts of approach followed by a retreat to a safe distance, presumably to balance exploration to discover possible rewards with the potential risk of predation. However, there is considerable heterogeneity in this exploratory behaviour, both across time as an individual subject becomes more confident in approaching the object, and across subjects; with some mice rapidly becoming confident to closely explore the object, while other timid mice never become fully confident that the object is safe. The current work aims to explain both the dynamics of adaptation of individual animals over time, and the quantitative and qualitative differences in behaviour between subjects, by modelling their behaviour as arising from model-based planning in a Bayes adaptive Markov Decision Process (BAMDP) framework, in which the subjects maintain and update probabilistic estimates of the uncertain hazard presented by the object, and rationally balance the potential reward from exploring the object with the potential risk of predation it presents.In order to fit these complex models to the behaviour the authors necessarily make substantial simplifying assumptions, including coarse-graining the exploratory behaviour into phases quantified by a set of summary statistics related to the approach bouts of the animal. Inter-individual variation between subjects is modelled both by differences in their prior beliefs about the possible hazard presented by the object and by differences in their risk preference, modelled using a conditional value at risk (CVaR) objective, which focuses the subject's evaluation on different quantiles of the expected distribution of outcomes. Interestingly these two conceptually different possible sources of inter-subject variation in brave vs timid exploratory behaviour turn out not to be dissociable in the current dataset as they can largely compensate for each other in their effects on the measured behaviour. Nonetheless, the modelling captures a wide range of quantitative and qualitative differences between subjects in the dynamics of how they explore the object, essentially through differences in how subject's beliefs about the potential risk and reward presented by the object evolve over the course of exploration, and are combined to drive behaviour.Exploration in the face of risk is a ubiquitous feature of the decision-making problem faced by organisms, with strong clinical relevance, yet remains poorly understood and under-studied, making this work a timely and welcome addition to the literature.Strengths:(1) Individual differences in exploratory behaviour are an interesting, important, and under-studied topic.(2) Application of cutting-edge modelling methods to a rich behavioural dataset, successfully accounting for diverse qualitative and qualitative features of the data in a normative framework.(3) Thoughtful discussion of the results in the context of prior literature.Limitations:(1) The model-fitting approach used of coarse-graining the behaviour into phases and fitting to their summary statistics may not be applicable to exploratory behaviours in more complex environments where coarse-graining is less straightforward.(2) Some aspects of the work could be more usefully clarified within the manuscript.

We thank reviewer #3 for their positive feedback and helping us to improve the clarity of our paper. We have added discussion they thought was missing.

**Reviewer #1 (Recommendations for the authors):**
(1) Line 25-28This part of the Abstract might give an impression that timidity (but not braveness) is potentially associated with psychiatric illness and even that timidity is thus inferior to braveness. However, even though extreme timidity might indeed be associated with anxiety or depression, extreme braveness could also be associated with other psychiatric or behavioral problems. Moreover, as a population, the existence of both timid and brave individuals could be advantageous, and it could be a reason why both types of individuals evolutionarily survived in the case of wild animals (although Akiti et al. used mice, which may have no or very limited genetic varieties, and so things may be different). So I would like to encourage the authors to elaborate on the expression of this part of the Abstract and/or enrich the related discussion in the Discussion.

This is an important point. We note on line 38 that excessive novelty seeking (potentially caused by excessive braveness) could also be maladaptive.

Additionally, we have added a paragraph to the discussion discussing heterogeneity in risk sensitivity within a population.

“Our data show that there is substantial variation in the degrees of risk sensitivity across the mice. Previous works have reported substantial interpopulation and intrapopulation differences in risk-sensitivity in humans which depend on gender, age, socioeconomic status, personality characteristics, wealth and culture (Rieger et al., 2015; Frey et al., 2017). Despite the normative appeal of 𝛼 = 1, it is possible that a population may benefit from including individuals with $\alpha$ different from 1.0 or highly negative priors. For example, more cautious individuals could learn from merely observing the risky behavior of less cautious individuals. Furthermore, we have only considered risk-sensitivity under epistemic uncertainty in our work. Risk averse individuals, for instance with 𝛼 < 1 may be more successful than risk-neutral agents in environments where there are unexpected dangers (unknown unknowns). Risk-aversion is thus a temperament of ecological and evolutionary significance (Réale et al., 2007).”

(2) Line 149Section 2.2 consists of eight subsections. I think this organization may not be very appealing, because there are a bit too many subsections, and their relations are not immediately clear to readers. So I would like to encourage the authors to make an elaboration. For example, since 2.2.1 - 2.2.5 describes a summary of model construction and model fitting whereas 2.2.6-2.2.8 shows the results, it could be good to divide these into separate sections (2.2.1 - 2.2.5 and 2.3.1 - 2.3.3).

Thank you for pointing this out. We’ve renumbered the sections as you’ve suggested.

(3) Line 347-8Theoretically, the effect of prior is diluted over experience whereas the effect of biased (risk-aversive) evaluation persists, as the authors mentioned in Lines 393-394. Then isn't it possible to consider environments/conditions in which the two effects can be separated?

We appreciate this suggestion. Indeed, our original thought in modeling this experiment was that this would be exactly the case here - with epistemic uncertainty reducing as the object became more familiar. However, proving to an animal that a single environment is completely stationary/fixed is hard - reflected in our conclusion here that the exploration bonus pool replenishes. Thus, we argued in the discussion that a series of environments would be necessary to separate risk sensitivity from priors.

(4) Line 407It would be nice to add a brief phrase explaining how (in what sense) this model's assumption was consistent with the reported behavior. Also, should the assumption of having two discrete approach states (cautious and confident) itself be regarded as a limitation of the model? If the tail-behind and tail-exposure approaches were not merely operationally categorized but were indicated to be two qualitatively distinct behaviors in the experiment by Akiti et al., it is reasonable to model them as two discrete states, but otherwise, the assumption of two discrete states would need to be mentioned as a simplification/limitation.

We have now removed line 407, and now have an additional paragraph in the discussion discussing the limitations of the tail-behind and tail-exposure state representation: “Motivated by tail-behind versus tail-exposed in Akiti et al. (2022), we model approach using a dichotomy between cautious and confident approach states. This is likely a crude approximation to the continuous and multifaceted nature of animal approach behavior. For example, during approach animals likely adjust their levels of vigilance continuously (or discretely; Lloyd and Dayan (2018)) to monitor threat, and choose different velocities for movement, and different attentional strategies for inspecting the novel object. We hope future works will model these additional behavioral complexities, perhaps with additional internal states, and corroborate these states with neurobiological data.”

(5) Line 418The authors contrasted their model-based analyses with the model-free analyses of Akiti et al. Another aspect of differences between the authors' model and the model of Akiti et al. is whether it is normative or mechanistic: while how the model of Akiti et al. can be biologically implemented appears to be clear (TS dopamine represents threat TD error, and TS dopamine-dependent cortico-striatal plasticity implements TD error-based update of model-free threat prediction), biological implementation of the authors' model seems more elusive. Given this, it might be a fruitful direction to explore how these two models can be integrated in the future.

We enthusiastically agree that it would be most interesting in the future to explore the integration of the two models - and, in the discussion (Lines 537-548, 454-461) , point to some first steps that might be fruitful along these lines. There are two separate considerations here: one is that our account is mostly computational and algorithmic, whereas Akiti’s model is mostly algorithmic and implementational; the second is, as noted by the reviewer, that our account is model-based, whereas Akiti’s model is model-free (in the sense of reinforcement learning; RL). These are related - thanks in no small part to the work from the group including Akiti, we know a lot more about the implementation of model-free than model-based RL. However, our model-based account does reach additional features of behavior not captured in Akiti et al.’s model such as bout duration, frequency, and approach type. Thus, the temptation of unification.

(6) Line 426Related to the previous point, it would be nice to more specifically describe what variable TS dopamine can represent in the authors' model if possible.

In the discussion (Lines 454-461) , we speculate that TS dopamine could still respond to the physical salience of the novel object and affect choices by determining the potential cost of the encountered threat or the prior on the hazard function. For example, perhaps ablating TS dopamine reduces the hazard priors which leads to faster transition from cautious to confident approach and longer bout durations, consistent with the optogenetics behavioral data reported in Akiti et al.

**Reviewer #2 (Recommendations for the authors):**
My guess is simpler versions of the model would not fit the data well. But this does not mean for example that the mice have probability distortions (CvaR) or that even probabilistic reasoning and the internal models necessary to support them are acting in the behavioral context studied by Akiti. So related to the above, I would ask what other models would fit and would not fit the data? And what does this mean?

These are good points. Our model provides an approximately normative account of the animals’ behavior in terms of what it achieves relative to a utility function. In practice, the animals could deploy a precompiled model-free policy (which does not rely on probabilistic computations) that is exactly equivalent to our model-based policy. With the current experiment, we cannot conclude whether or not the animals are performing the prospective calculations in an online manner. Of course, the extent to which animals or humans are performing probabilistic computations online and have internal models are on-going questions of study.

Model comparison is difficult because currently we do not know of any other risk-sensitive exploration models. We cannot directly compare to the model in Akiti et al. since our model explains additional features of behavior: bout duration, frequency, and approach type. Indeed, our model is as simple as it can be in the sense with the exception of nCVaR, removing any of the other parameters makes it difficult to fit some animals in our dataset. In the future, our model could be used to fit other datasets of risk-sensitive exploration and, ideally, be compared to other models.

Explaining why animals avoid the novel object in what the offers call benign environment is a very tricky issue. In Akiti et al, the readers are not yet convinced that the mice know that this environment is benign. Being placed in an arena with a novel object presents mice with a great uncertainty and we do not know whether they treat this as benign. Therefore, the alternative explanations in this study need to be carefully discussed in lieu of the limitations of the initial study.

It is certainly true that it is unclear if the arena is completely benign to the animals. However, the amount of time the animal spends in the center of the arena decreases significantly from habituation to novelty days. This suggests that the animals avoid the novel object largely because of the object itself, rather than the potential danger associated with the arena. Furthermore, the animals are not reported as exhibiting more extreme behaviours such as freezing. In any case, our account is relative in the sense that we are comparing the time the animal spends at the object versus elsewhere in the environment, driven by the relative novelty and relative risk of the environment versus the object. Trying to get more absolute measures of these quantities would require a richer experimental set-up, for instance with different degree of habituation or experience of the occurrence of (other) novel objects, in general.

We added a short note to the discussion to explain this:

“Fourth, we modeled the relative amount of time the animal spends at the object versus elsewhere in the environment which depends on the differential risk in the two states. However, it is likely the animals avoid the novel object largely because of the object itself, rather than the potential danger associated with the arena since they spend much less time at the center of the arena during novelty than habituation days.”

Figure 2 - how confident are the authors that each mouse differs from y=1? Related to this, the behavior in Akiti is very noisy and changes across time. I am not sure if the authors fully describe at what levels their model captures the behavior vs not in a detailed enough fashion.

We have performed a random permutation test on the minute-to-minute data. We have updated Figure 2 so that brave animals that pass the Benjamini–Hochberg procedure y>1 at level q=0.05 are represented with solid green dots and animals that don’t pass are represented with hollow dots. 8 out of 11 brave animals passed Benjamini–Hochberg.

**Reviewer #3 (Recommendations for the authors):**
(1) I could not find information in the preprint about code availability. Please consider making the code public to help others apply these modelling methods.

We have released code and included the url in the paper in the Methods section.

(2) Though the manuscript was generally clearly written, there were a number of places where some additional information or clarification would be useful:a) Please define and explain the terms 'tail-behind' and 'tail-exposed' (used to describe approach bout types) when first used.

We have added definitions when we first mention these terms:

“[...] 'tail-behind' (bouts where the animal's nose was closer to the object than the tail for the entire bout) and 'tail-exposed' (bouts where the animal's tail is closer to the object than the nose at some point during the bout), associated respectively with cautious risk-assessment and engagement”

b) At lines 57-58 when contrasting the 'model-free' account of Akiti et al with the 'model-based' account of the current work, it would be worth clarifying that these terms are being used in the RL sense rather than e.g. a model-based analysis of the data.

We have updated the relevant lines to say “model-free/based reinforcement learning”.

c) Line 61, the phrase 'the significant long-run approach of timid animals despite having reached the "avoid" state' is unclear as the 'avoid' state has not been defined.

We updated the terminology to “avoidance behavior” to be consistent with Akiti et al. Avoidance refers to the animal routinely avoiding the object and therefore being unable to learn whether it is safe.

d) It was not completely clear to me how the coarse-graining of the behaviour was implemented. Specifically, how were animals assigned to the brave, intermediate, or timid group, and how were the parameters of the resulting behavioural phases fit?

Sorry that this was not clear. Section 2.1 explains how the minute-to-minute behavioral data was coarse-grained and how animal groups were assigned. We have added further explanation of Figure 2 to the main text:

“Fig 2 summarizes our categorization of the animals into the three groups: brave, intermediate, and timid based on the phases identified in the animal's exploratory trajectories. Timid animals spend no time in confident approach and are plotted in orange at the origin of Fig 2. Brave animals differ from intermediate animals in that their approach time during the first ten minutes of the confident phase is greater than the last ten minutes (steady-state phase). Brave animals are plotted in green above and intermediate animals are plotted in black below the y=1 line in Fig 2.”

We also added extra information to outline the goal, and methodology of coarse-graining and animal grouping:

“We sought to capture these qualitative differences (cautious versus confident) as well as aspects of the quantitative changes in bout durations and frequencies as the animal learns about their environment. To make this readily possible, we abstracted the data in two ways:

averaging bout statistics over time, and clustering the animals into three groups with operationally distinct behaviors.”

e) What purpose does the 'retreat' state serve in the BAMDP model (as opposed to transitioning directly from 'object' to 'nest' states), and why do subjects not pass through it following 'detect' states?

Thank you for pointing this out. We have updated Figure 3 to note that the two “detected states” also point to the “retreat” state. The reviewer is correct that there could be alternative versions of the state diagram, and the ‘retreat’ state could indeed have been eliminated. However, we thought that it was helpful to structure the animal’s progress through state space.

f) Why was the hazard function parameterised via the mean and SD at each time step rather than with a parametric form of the mean and SD as a function of time?

Since the agent can only spend 2, 3, or 4 turns at the object states, we didn’t see a need to parameterize the mean and SD as a function of time. Doing so is a good solution to scaling up the hazard function to more time-steps.

(3) There were also a couple of points that could potentially be usefully touched on in the discussion:a) What, if any, is the relationship between the CVaR objective and distributional RL? They seem potentially related due to both focussing on quantiles of the outcome distribution.

We have added a paragraph to the discussion discussing the connection between distributional RL and CVaR:

“CVaR is known to come in different flavors in the case of temporally-extended behavior. Gagne and Dayan (2021) introduces two alternative time-consistent formulations of CVaR: nested CVaR (nCVaR) and precommitted CVaR (pCVaR). nCVaR and pCVaR both enjoy Bellman equations which make it possible to compute approximately optimal policies without directly computing whole distributions of the outcomes. We use nCVaR in this study for its computational efficiency. There is, of course, great current interest in distributional reinforcement learning (Bellemare et al., 2023b) which does acquire such whole distributions, not the least because of prominent observations linking non-linearities in the response functions of dopamine neurons to methods for learning distributions of outcomes (Dabney et al., 2020; Masset et al., 2023; Sousa et al., 2023). One functional motivation for considering entire outcome distributions is the possibility of using them to determine risk-sensitive policies (Gagne and Dayan, 2021).

While it is possible to compute CVaR directly from return distributions, Gagne and Dayan (2021) showed that this can lead to temporally inconsistent policies where the agent deviates from its original plans (the authors called this the fixed CVaR or fCVaR measure).

Rather further removed from our model-based methods is work from Antonov and Dayan (2023), who consider a model-free exploration strategy which exploits full return distributions to compute the value of perfect information which is used as a heuristic for trying actions with uncertain consequences. Future works can examine risk-sensitive versions of Antonov and Dayan (2023)'s computationally efficient model-free algorithm as one solution to the burdensome computations in our model-based method.”

b) Why normatively might subjects have non-neutral risk preference as captured by the CvaR?

We also added a paragraph to the discussion discussing the advantage of heterogeneity in risk sensitivity within a population:

(Reviewer #1 had the same question, see above) “Our data show that there is substantial variation in the degrees of risk sensitivity across the mice. Previous works have reported substantial interpopulation and intrapopulation differences in risk-sensitivity in humans which depend on gender, age, socioeconomic status, personality characteristics, wealth and culture [...]”

c) Relevance of the current modelling work to clinical conditions characterised by dysregulation of risk assesment (e.g. anxiety or PTSD).

We’ve added a paragraph to the discussion:

“Inter-individual differences in risk sensitivity are also of critical importance in psychiatry, reflected in a panoply of anxiety disorders (Butler and Mathews, 1983; Giorgetta et al., 2012; Maner et al., 2007; Charpentier et al., 2017), along with worry and rumination (Gagne and Dayan, 2022). Understanding the spectrum of extreme priors and extreme values of 𝛼 could have therapeutic implications, adding significance to the search for tasks that can more cleanly separate them.”

d) Is it surprising to see differences in risk preference (nCVaR) between the familiar object and novel object condition, given that risk preference might be conceptualised as a trait rather than a state variable?

Thank you for raising this point. You are right that we expected risk sensitivity (nCVaR alpha) to be the same between FONC and UONC animals on average. It is difficult to know if alpha is higher for FONC than UONC animals due to the non-identifiability between alpha and hazard priors. We have added this discussion to the paper:

“This is surprising if we interpret 𝛼 as a trait that is stable through time. Unfortunately, due to the non-identifiability between 𝛼 and hazard priors, we cannot verify whether 𝛼 is actually higher for FONC animals than UONC animals.”